# Development of an effective illness severity measure and assessment of the impact of perceived illness severity on formal careseeking for fatal illnesses of neonates and infants in six sub-Saharan Africa countries and Pakistan

Henry D. Kalter[1]*, Jamie Perin[1,2], Zulfiqar A. Bhutta[3,4], Inuwa B. Jalingo[5], Paul R. Libite[6], Abdou Maina[7], Tiope Mleme[8], Mlemba A. Kamwe[9], Ivalda Macicame[10], Robert E. Black[1]

1 Institute for International Programs, Department of International Health, Johns Hopkins Bloomberg School of Public Health, Baltimore, Maryland, United States of America, 2 Center for Child and Community Health Research, Department of Pediatrics, Johns Hopkins School of Medicine, Baltimore, Maryland, United States of America, 3 Institute for Global Health & Development, Aga Khan University, Karachi, Pakistan, 4 SickKids Centre for Global Child Health, Hospital for Sick Children, Toronto, Ontario, Canada, 5 Census Department, National Population Commission, Abuja, Nigeria, 6 Institut National de la Statistique, Yaoundé, Cameroun, 7 Institut National de la Statistique, Niamey, Niger, 8 National Statistical Office, Zomba, Malawi, 9 National Bureau of Statistics, Dodoma, United Republic of Tanzania, 10 Instituto Nacional de Saúde, Maputo, Moçambique

* hkalter1@jhu.edu

## Abstract

Early careseeking for sick children can make the difference between life and death. Verbal autopsy (VA) studies of the cause of death typically ask about severe symptoms such as seizures and possibly mild or moderate symptoms such as rash, but without examining the relationship between caregivers' perception of illness severity and appropriate careseeking. Verbal and social autopsy (VASA) is a newer method that builds on VA by also examining social factors related to death. From seven VASA studies conducted in Africa and Asia we developed a 2-sign method based on activity level and feeding behavior and a multiple sign method of identifying mild, moderate and severe illness of neonates and 1–11-month-olds. We then examined the relationship of caregivers' perception of their child's condition at illness onset and several covariates to seeking formal health care during the fatal illness. The 2-sign and multiple sign methods effectively distinguished mild, moderate, and severe illnesses, respectively, of neonates and 1–11-month-olds. Careseeking was almost uniformly decreased for severely ill neonates (8.4%-41.8% vs mild: 15.0%-66.7% and moderate: 30.5%-68.5%, p=0.12-<0.001), but multivariate analysis revealed that older age in all six African countries (AOR 1.11 [95% CI 1.02, 1.21], p=0.02 to 1.10 [1.04, 1.16], p<0.001) and moderate illness in three (4.83 [1.06, 21.96], p=0.04 to 4.35 [1.59, 11.93], p=0.005) were associated with careseeking, while severe illness was no

**Data availability statement:** The data utilized for the analyses conducted for this paper all come from de-identified minimal subsets of the full data collected by prior verbal and social autopsy studies in seven countries. Five of the minimal datasets, from Nigeria, Niger, Malawi, Tanzania and Mozambique, have been provided to the Journal of Global Public Health as Supporting Information files and will be uploaded to the figshare repository for open access upon publication of this paper. The other two countries, Cameroon and Pakistan, require that data requestors contact a country representative to request their minimal dataset and permission to utilize the data. To receive the Cameroon data, requestors should contact the Director General of the National Institute of Statistics at PO Box 134, Yaoundé, Cameroon, email: infos@ins-cameroun.cm to request the data and permission to use the data for the purpose of writing and publishing a scientific article(s). Furthermore, data requestors must provide a copy of the final version of any published article(s) utilizing the data to the Cameroon NIS at the same address and must inform the Director General of any other uses that will be made of the data. As no standard data use agreement form is currently available for this purpose, any person or institution requesting this data should formulate their request to the NIS in their own way, including their commitment to use the data according to the rules defined above. To receive the Pakistan data, requestors should contact Dr. Salman Kirmani, Director CoE-WCH at salman.kirmani@aku.edu. A data use agreement on a specific Aga Khan University template will be provided and must be signed by requestors to receive the data. Any published article(s) utilizing all or part of the minimal dataset of any of the seven countries must include the respective same acknowledgement for use of the data from that country(ies) found in the Acknowledgements section of the current publication, as well as acknowledging the current publication as the source of the minimal dataset utilized by the data requestor.

**Funding:** Work that contributed toward the current analysis was conducted for the Improving

longer significant. Similar to neonates, older age in three of five countries (1.26 [1.01, 1.58], p = 0.046 to 1.10 [1.03, 1.16], p = 0.003) and moderate illness in one (2.24 [1.17, 4.30], p = 0.016) were drivers of careseeking for 1–11-month-olds. Careseeking was increased in some countries for infectious illnesses but not for intrapartum- or prematurity-related conditions. Child mortality studies should assess severity level and caregivers' response at various illness stages. Because older infants have more specific illness signs, the 2-sign method should be used only for neonates. Behavior change messages encouraging careseeking for moderate illness signs should be developed. The 2-sign method can serve as a practical tool for this purpose for illnesses of neonates. Effective interventions may require overcoming local barriers to careseeking and bringing delivery and newborn care closer to communities to prevent and treat early onset neonatal illnesses.

## Introduction

Caregivers' timely seeking of appropriate health care for their children's potentially fatal illnesses can make the difference between life and death. Study of the interplay of recognition of illness signs, perception of disease severity, and responsive careseeking might address this issue and provide fruitful insights into the development of effective interventions. Verbal autopsy (VA) is an interview-based methodology used for several years to determine cause of death in low resource settings where many or most child deaths occur outside of medical care. However, VA questionnaires typically have focused purely on identifying illness signs and symptoms, including severe symptoms such as severe cough and inability to suckle, and possibly mild or moderate symptoms such as rash and fever, but without any reference to careseeking and minimal inquiry about the timing of symptom occurrence. Only recently has the most prominent VA instrument included an open-ended narrative section that asks about respondent recognition of 'symptoms' and careseeking for the fatal illness [1].

A methodology that builds on and extends VA, called verbal and social autopsy (VASA), has addressed this issue by examining the timing of careseeking behavior relative to caregivers' perception of the severity of their child's illness [2–4]. A scale across severity levels, especially one based on signs and symptoms easily recognized by and of concern to child caregivers [5,6] and shown to have objective validity, such as activity level and feeding behavior [7–9], would be a boon to studies that aim to assess the contribution of illness perception and delayed careseeking to child mortality.

Such a tool could also contribute to the development of behavior change messages based on illness signs that should prompt careseeking before severe illness develops [10]. Some researchers have found that caregivers mainly respond to severe symptoms [11,12] or even fail to perceive the severity of their child's illness [11] and that this leads to delayed careseeking and lessened chance for survival once reaching a health care provider [11,12]. However, others have found that mothers do respond to less severe symptoms [13], indicating the potential utility of such a tool.

Measurement and Program Design (IMPROVE) project funded by the Bill and Melinda Gates Foundation (BMGF) under grant OPP1172551 to Johns Hopkins University (JHU). The VASA studies in Cameroon and Malawi were funded by BMGF through a grant to the U.S. Fund for UNICEF for the Child Health Epidemiology Reference Group (CHERG) project of JHU. The Niger VASA study was funded by the same CHERG grant and the UNICEF country office of Niger. The VASA studies in Tanzania and Pakistan were funded by a grant from BMGF to the Maternal and Child Epidemiology Estimation (MCEE) project under Global Development Grant #OPP1096225 to JHU. The VASA study in Pakistan was supported by additional funding from BMGF through the IMPROVE project grant. Funding for the VASA study field work in Nigeria was provided by the U.S. Agency for International Development/Nigeria and the Office of Population and Reproductive Health, U.S. Agency for International Development/ Washington D.C. through a Leader with Associates (LWA) Cooperative Agreement under the terms of Award No GHSA-00-09-00004-00 to JHU. Technical support for the study was provided by JHU through the MCEE project funded by BMGF. The VASA study in Mozambique was funded by BMGF grant number OPP1163221 to JHU. No funder had a role in study design, data collection and analysis, decision to publish, or preparation of the manuscript. The authors' views expressed in this publication do not necessarily reflect the views, nor those of and/or the decisions, policy, or views of the funders.

**Competing interests:** The authors have declared that no competing interests exist.

The current paper develops sign-based methods of identifying mild, moderate, and severe conditions of the most vulnerable children, including neonates and 1–11-month-old infants, at the onset of a fatal illness. It then uses these methods to assess the relationship of caregivers' perception of their child's illness severity level to other variables affecting child mortality and the contribution of these factors to seeking formal health care for their sick neonates and infants.

## Materials and methods

This study was a secondary analysis of data from five national VASA studies conducted in Niger [2], Tanzania [3], Nigeria [14], Mozambique [15] and Pakistan, and three- and two-district VASA studies, respectively, in Cameroon (Doume, Nguele-mendouka and Abong-Mbang districts) [16] and Malawi (Balaka and Salima districts) [17]. The Mozambique data were collected as part of a prospective national sample registration system with VASA, while the data for all other countries are from cross-sectional retrospective studies of deaths identified by a household survey (Niger, Tanzania, Nigeria, Malawi, Pakistan) or census (Cameroon). The years of the deaths/ conduct of VASA interviews (mean, median years interview recall period) in each country were: Cameroon 2006–10/ 2012 (3.2, 3), Nigeria 2008–13/ 2014 (3.7, 4), Malawi 2008–12/ 2013 (3.4, 3), Niger 2007–10/ 2012 (3.3, 3), Tanzania 2010–15/ 2017 (4.2, 4), Mozambique 2018–20/ 2018–20 (0.2, 0), and Pakistan 2015–19/ 2018–19 (1.5, 1). In five countries 13% to 42% of interviews were conducted more than four years after the death, while in Pakistan and Mozambique, respectively, 97% were conducted in less than four years and 100% in less than two years. The deceased child's mother was the respondent for 77%, 96%, 87%, 94%, 93%, 86% and 91% of VASA interviews, respectively, in Cameroon, Nigeria, Malawi, Niger, Tanzania, Mozambique and Pakistan.

The VASA questionnaire used in all the African countries except Mozambique combined the Population Health Metrics Research Consortium (PHMRC) VA questionnaire [18] with the original Child Health Epidemiology Reference Group (CHERG) SA questionnaire [19]; while the VASA instrument in Pakistan and Mozambique incorporated the 2016 WHO VA questionnaire and an updated version of the CHERG SA questionnaire [20]. Survey weights were applied to the data from Malawi, Niger, Nigeria, Tanzania and Mozambique and all analyses were conducted such as to account for the multi-stage sampling designs of the platform surveys. Weights were not needed in Pakistan and neither weights nor correcting for the sampling design were required in Cameroon, where the deaths were identified through a population census of the study districts.

### Illness severity classification

We developed three methods using VA-identified signs and symptoms to classify illness severity during various stages of neonates' and infants' fatal illnesses; and compared the performance of the first method with the second and third methods in distinguishing mild, moderate and severe illness when first noted by the child's

caregiver (method 1) or on illness day-1 (methods 2 and 3). This comparison consisted of cross tabulating the method 1 findings in each country with those of methods 2 and 3, which utilize many more illness signs that method 1, and by examining the mean number of methods 2 and 3 urgent referral/severe illness signs corresponding with the method 1 mild, moderate, and severe illness classifications.

The first method consists of a simple 2-sign severity score based on reported activity level and feeding behavior (Panel 1). Caregivers were asked about these signs when they first noted the child was ill (defined as at onset), when it was decided to seek formal health care for the illness, and when the child left the first health provider alive. As a sensitivity test of whether the method is capable of identifying severe illness, hypothesizing that neonates who died from early onset illnesses such as perinatal asphyxia and very low birth weight would be more severely ill at onset, we compared the 2-sign severity scores of all neonates to those who died at age 0 or 1 day.

**Panel 1. 2-sign method of classifying illness severity**

| Individual Parameters | Levels and assigned scores | | |
|---|---|---|---|
| | Normal = 1 | Medium = 2 | Abnormal = 3 |
| Feeding | Normally | Poorly | Not at all |
| Activity | Normal | Less active | Not moving |

**Combined scores and final analysis ranks**

| Illness level | Signs | Score | Rank |
|---|---|---|---|
| Mild | 2 normal | 2 | 1 |
| | or 1 normal and 1 medium | 3 | 1 |
| Moderate | 1 normal and 1 abnormal | 4 | 2 |
| | or 2 medium | 4 | 2 |
| Severe | 1 medium and 1 abnormal | 5 | 3 |
| | or 2 abnormal | 6 | 3 |

The second method examines for the presence of one or more of 15 neonatal and 13 child WHO/UNICEF Integrated Management of Childhood Illness (IMCI) signs of illness and severe illness requiring referral [21,22]; and the third method adds 18 neonatal signs and 16 child signs found only in the VASA formats (Panel 2). Caregivers were asked on which illness day, with the day of illness onset defined as day-1, each reported sign was first noted. For both methods 2 and 3, the absence of all the signs is classified as a mild illness, the presence only of other illness sign(s) identifies a moderate illness, and any urgent referral/severe illness sign(s) indicates a severe illness. Comparisons of the methods were not possible in Pakistan and Mozambique because the VASA questionnaire utilized in those countries did not collect the illness day of onset for each reported illness sign and symptom.

**Panel 2. Integrated Management of Childhood Illness (IMCI) signs and Verbal and Social Autopsy (VASA) additional illness signs**

**Neonates (0–27 days old)\***

| IMCI signs requiring urgent referral | VASA additional signs of severe illness |
|---|---|
| Not able to feed | Not able to breathe immediately after birth |
| No movement at all | First cried >5 minutes after birth or never cried |
| Spasms or convulsions | Stopped being able to cry |
| Fast breathing and age at death = 0–6 days | Not able to open mouth when stopped suckling |
| Chest indrawing | Ulcers/pits |
| Fever | Areas of skin turned black |
| Cold to touch | Bleeding from anywhere |
| Not able to suckle normally on first day of life | Vomits everything |
| Stopped being able to suckle normally | Grunting |
| Yellow skin or eyes and age at death = 0 days | Bulging fontanelle |
| | Umbilical cord stump redness extending to skin |
| | Unresponsive/unconscious |

| IMCI other illness signs€ | VASA additional other illness signs |
|---|---|
| Feeding poorly | Lethargic |
| Pus drainage from umbilical cord stump | Bruises/signs of injury at birth |
| Redness of umbilical cord stump | Difficult breathing at birth |
| Skin bumps containing pus or a single large area with pus | First cried within 5 minutes after birth (did not cry immediately) |
| Fast breathing and age at death = 7–27 days | 4 or more loose/watery stools on worst day |
| | Difficult breathing |

| Infants (1–11 months old)* | |
|---|---|
| IMCI signs requiring urgent referral** | VASA additional signs of severe illness |
| Not able to feed | Severe cough |
| Convulsions | Vomiting after coughing |
| Unconscious | Grunting |
| Stridor | Wheezing |
| Stiff neck | Bulging fontanelle (in <18-month-olds) |
| Limbs became very thin | Skin flaked off in patches |
| Swollen legs or feet | Hair changed to reddish or yellowish color |
| | Bleeding from anywhere |
| | Areas of skin turned black |
| | Injury (road traffic, fall, drowning, poisoning, bite/sting of venomous animal, burn, violence, other injury) |

| IMCI other illness signsβ | VASA additional other illness signs |
|---|---|
| Feeding poorly | 3 or more loose/watery stools on the worst day |
| Fever | Cough for 7 or more days |
| Visible blood in loose or liquid stools | Difficult breathing |
| Chest indrawing | Blisters with clear fluid |
| Fast breathing | Swelling in the armpits |
| Pallor | Whitish rash in the mouth/on the tongue |

*The actual IMCI age categories are 'up to 2 months' and '2 months up to 5 years;' The current study utilizes the IMCI 'up to 2 months' signs to characterize neonatal illnesses, and the '2 months up to 5 years' signs for 1–11-month-old illnesses; €IMCI sign 'Movement only when stimulated' has no direct equivalent in the VASA questionnaire so is not included here. VASA's 'lethargic' is similar in meaning and is included as a VASA additional other illness sign. **IMCI signs 'Severe persistent diarrhea with dehydration' and 'Diarrhea with severe dehydration' cannot be identified using the VASA questionnaire due to missing questions for dehydration; βIMCI signs 'Diarrhea with some dehydration' and 'Persistent diarrhea without dehydration' cannot be identified using the VASA questionnaire due to missing questions for dehydration

## Illness recognition and careseeking

The better-performing method in each age group was then used to assess caregivers' perception of the child's condition when illness that started in the community, i.e., for which the child's caregiver could respond, was first noted or on illness day-1; and to examine its univariate associations with the child's age at death and first seeking formal health care, defined as care provided by a health professional (doctor, nurse, trained health worker) in a hospital, health center, health post or private office, or by a trained community health worker, nurse or midwife. (Table A in S1 Appendix shows the percentage of neonatal and 1–11-month-old deaths whose illness started in the community, as opposed to illness that started in the delivery facility.) Multivariable logistic regression models were developed to examine factors associated with seeking formal health care (yes, no). Possible predictor variables included illness severity at onset or on illness day-1 (mild, moderate, severe), age at death (neonates: days; 1–11-month-olds: months), hours to reach the usually visited health facility in an emergency, the child's mother's age (years), the child's mother's completed years of formal schooling, and cause of death as determined by the Expert Algorithm (EAVA) analysis method [23] (neonates: severe infection, intrapartum related events [IPRE = birth asphyxia or injury] or prematurity, all other causes [reference value]; 1–11-month-olds: severe febrile infection, all other causes [reference value]). Neonatal severe infection included meningitis, pneumonia, and sepsis; 1–11-month-old severe febrile infection included AIDS, measles,

meningitis, pertussis, pneumonia, malaria, hemorrhagic fever, and fever with rash, convulsions, or unconsciousness. We included diarrhea in the comparison group because it frequently presents without fever and formal careseeking for childhood diarrhea has been found to be strongly associated with fever [24]. SAS version 9.4 [25] was used to conduct all statistical analyses.

All analyses excluded cases missing caregiver's perception of illness severity, as the primary variable of interest, or the outcome variable formal careseeking. Complete case (CC) multivariable analyses were conducted that also excluded cases missing any of the other predictor variables. Multivariable analyses were also done that substituted imputed mean (IM) values for missing travel time, mother's age and mother's schooling. There were no missing values for cause of death and child's age. Complete case analyses are presented in the main paper for countries with less than 10% loss of data points due to missingness in at least one factor; otherwise, analyses using imputed means are presented [26].

### Ethics statement

Ethical clearance for the VASA study in each country was obtained from the Institutional Review Board of the Johns Hopkins Bloomberg School of Public Health (Cameroon: IRB No: 3868, Nigeria: IRB No: 5163, Malawi: IRB No: 2247, Niger: IRB No: 3274, Tanzania: IRB No: 7820, Mozambique: IRB No: 7867, Pakistan: IRB No: 8688) and the national ethics committee in each country, including the Cameroon National Ethics Committee (AUTORISATION N°285/CNE/SE/2011), the National Health Research Ethics Committee of the Nigeria Federal Ministry of Health (Approval Number NHREC/01/01/2007-30/10/2013), the Malawi National Health Sciences Research Committee (Approval number 927), the National Consultative Ethics Committee of the Niger Ministry of Health (DELIBERATION N0 014/2011/CCNE), the Tanzania National Institute for Medical Research (NIMR/HQ/R.8a/Vol.IX/2550) and the Ministry of Health of Zanzibar (PROTOCOL NO. ZAMREC/0001/JULY/17), the Mozambique National Health Bioethics Committee (REF 608/CNBS/17), and in Pakistan, the National Bioethics Committee (Ref: No.4-87/NBC-304/18/1026) and the Aga Khan University Ethics Review Committee (2018-0444-461). All respondents provided informed consent, either oral or written, depending on the country's requirement and anticipated literacy of the respondents, before the VASA interview was conducted. Ethical review for the current analysis was not sought because all data are from the previously approved studies and were either de-identified early in the conduct of those studies or personal identifiers were encrypted at the time of data collection and indiscernible by the past and present studies' authors. The data were accessed for the current analysis from February 11, 2023 – February 22, 2025.

## Results

### Study population

The full sample sizes of neonates in the seven countries ranged from 164 in Cameroon to 2,088 in Pakistan, with those whose illness started at home ranging from 115 in Tanzania to 1,381 in Pakistan. The number of 1–11-month-olds ranged from 158 in Tanzania to 691 in Nigeria (Table A in S1 Appendix).

Tables 1 a and 1b present some demographic characteristics of the study population. Two-thirds to 85% of neonatal deaths were of early neonates, while infant deaths averaged about six months of age. Table 4 provides additional information on neonates' and 1–11-month-olds' age relative to their illness severity at onset. Males predominated among neonatal deaths, but infant deaths were nearly equally distributed among males and females.

Home births and deaths were common, most markedly in Nigeria and Niger, with nearly 80% of deaths in Niger occurring at home. This appeared to correlate with a lack of maternal education, with nearly half to about 85% of mothers in these countries having received no schooling. However, there did not seem to be an association with mother's young age,

**Table 1. a. Selected demographic characteristics of neonates in seven study countries. b. Selected demographic characteristics of 1-11-month-olds in five study countries.**

| Characteristic | Cameroon | Nigeria | Malawi | Niger | Tanzania | Mozambique | Pakistan |
|---|---|---|---|---|---|---|---|
| | N (%) | N (%) | N (%) | N (%) | N (%) | N (%) | N (%) |
| **Child's age (days)** | | | | | | | |
| 0-6 | 116 (70.7) | 535 (74.0) | 224 (70.0) | 311 (68.7) | 195 (85.5) | 302 (74.9) | 1,629 (78.0) |
| 7-27 | 48 (29.3) | 188 (26.0) | 96 (30.0) | 142 (31.3) | 33 (14.5) | 101 (25.1) | 459 (22.0) |
| μ (SE) days | 4.6 (0.7) | 4.5 (0.2) | 5.1 (0.4) | 6.1 (0.3) | 3.1 (0.4) | 4.5 (0.3) | 4.3 (0.1) |
| **Sex** | | | | | | | |
| Male | 96 (58.5) | 421 (58.2) | 183 (57.2) | 264 (58.3) | 136 (59.6) | 210 (54.8) | 1282 (61.4) |
| Female | 68 (41.5) | 301 (41.6) | 137 (42.8) | 189 (41.7) | 92 (40.4) | 173 (45.2) | 806 (38.6) |
| Don't know | 0 (0) | 1 (0.1) | 0 (0) | 0 (0) | 0 (0) | 20 (5.0) | 0 (0) |
| **Birthplace** | | | | | | | |
| Home | 92 (56.1) | 418 (57.8) | 81 (25.3) | 313 (69.1) | 53 (23.2) | 163 (40.4) | 619 (29.6) |
| Hospital | 52 (31.7) | 172 (23.8) | 119 (37.2) | 26 (5.7) | 109 (47.8) | 107 (26.6) | 1,165 (55.8) |
| Other facility | 12 (7.3) | 78 (10.8) | 96 (30.0) | 102 (22.5) | 51 (22.4) | 103 (25.6) | 283 (13.6) |
| Other | 8 (4.9) | 55 (7.6) | 24 (7.5) | 12 (2.6) | 15 (6.6) | 30 (7.4) | 18 (0.9) |
| Don't know | 0 (0) | 0 (0) | 0 (0) | 0 (0) | 0 (0) | 0 (0) | 3 (0.1) |
| **Place of death** | | | | | | | |
| Home | 93 (56.7) | 478 (66.1) | 122 (38.1) | 349 (77.0) | 68 (29.8) | 221 (54.8) | 799 (38.3) |
| Hospital | 50 (30.5) | 140 (19.4) | 119 (37.2) | 29 (6.4) | 109 (47.8) | 123 (30.5) | 1,154 (55.3) |
| Other facility | 5 (3.0) | 55 (7.6) | 55 (17.2) | 51 (11.3) | 35 (15.4) | 32 (7.9) | 48 (2.3) |
| Other | 16 (9.8) | 49 (6.8) | 24 (7.5) | 24 (5.3) | 16 (7.0) | 27 (6.7) | 86 (4.1) |
| Don't know | 0 (0) | 1 (0.1) | 0 (0) | 0 (0) | 0 (0) | 0 (0) | 1 (0.05) |
| **Mother's age** | | | | | | | |
| 10-Dec | 6 (3.7) | 0 (0) | 0 (0) | 0 (0) | 0 (0) | 0 (0) | 0 (0) |
| 13-19 | 61 (37.2) | 150 (20.7) | 83 (25.9) | 96 (21.2) | 52 (22.8) | 127 (31.5) | 276 (13.2) |
| 20-34 | 75 (45.7) | 464 (64.2) | 196 (61.3) | 275 (60.7) | 133 (58.3) | 228 (56.6) | 1,537 (73.6) |
| 35+ | 17 (10.4) | 106 (14.7) | 34 (10.6) | 63 (13.9) | 41 (18.0) | 45 (11.2) | 246 (11.8) |
| Missing | 5 (3.0) | 3 (0.4) | 7 (2.2) | 19 (4.2) | 2 (0.9) | 3 (0.7) | 29 (1.4) |
| μ (SE) years | 22.7 (0.6) | 26.5 (0.3) | 24.5 (0.4) | 25.5 (0.3) | 26.2 (0.5) | 24.7 (0.4) | 26.5 (0.1) |
| **Mother's education** | | | | | | | |
| None | 4 (2.4) | 350 (48.4) | 46 (14.4) | 382 (84.3) | 30 (13.2) | 7 (1.7) | 2 (0.1) |
| Primary | 108 (65.9) | 168 (23.2) | 181 (56.6) | 49 (10.8) | 39 (17.1) | 198 (49.1) | 329 (15.8) |
| Secondary | 50 (30.5) | 162 (22.4) | 91 (28.4) | 14 (3.1) | 151 (66.2) | 90 (22.3) | 358 (17.1) |
| Higher | 2 (1.2) | 28 (3.9) | 1 (0.3) | 1 (0.2) | 8 (3.5) | 0 (0) | 72 (3.4) |
| Missing | 0 (0) | 15 (2.1) | 1 (0.3) | 7 (1.5) | 0 (0) | 108 (26.8) | 1,327 (63.6) |
| **HH* electricity** | | | | | | | |
| Yes | 42 (25.6) | 315 (43.6) | 11 (3.4) | 39 (8.6) | 49 (21.5) | 82 (20.3) | 1,983 (95.0) |
| No | 122 (74.4) | 408 (56.4) | 309 (96.6) | 414 (91.4) | 179 (78.5) | 321 (79.7) | 105 (5.0) |
| **HH* floor material** | | | | | | | |
| Mud/clay | 132 (80.5) | 246 (34.0) | 290 (90.6) | 416 (91.8) | 134 (58.8) | 320 (79.4) | 898 (43.0) |
| W/C/T** | 32 (19.5) | 442 (61.1) | 29 (9.1) | 36 (7.9) | 94 (41.2) | 83 (20.6) | 1,159 (55.0) |
| Other | 0 (0) | 35 (4.8) | 1 (0.3) | 1 (0.2) | 0 (0) | 0 (0) | 30 (1.4) |
| Don't know | 0 (0) | 0 (0) | 0 (0) | 0 (0) | 0 (0) | 0 (0) | 1 (0.05) |

*(Continued)*

Table 1. (Continued)

| Characteristic | Cameroon | Nigeria | Malawi | Niger | Tanzania | Mozambique | Pakistan |
|---|---|---|---|---|---|---|---|
| | N (%) | N (%) | N (%) | N (%) | N (%) | N (%) | N (%) |
| **Child's age (months)** | | | | | | | |
| 1-2 | 33 (12.6) | 159 (23.0) | 69 (20.6) | 60 (22.3) | 41 (25.9) | | |
| 3-6 | 96 (36.8) | 228 (33.0) | 115 (34.3) | 91 (33.8) | 50 (31.6) | | |
| 7-11 | 132 (50.6) | 304 (44.0) | 151 (45.1) | 118 (43.9) | 67 (42.4) | | |
| μ (SE) months | 6.5 (0.2) | 5.8 (0.1) | 5.8 (0.2) | 5.6 (0.2) | 5.5 (0.3) | | |
| **Sex** | | | | | | | |
| Male | 136 (52.1) | 350 (50.7) | 166 (49.6) | 137 (50.9) | 81 (51.3) | | |
| Female | 125 (47.9) | 341 (49.3) | 169 (50.4) | 132 (49.1) | 77 (48.7) | | |
| **Birthplace** | | | | | | | |
| Home | 180 (69.0) | 431 (62.4) | 54 (16.1) | 195 (72.5) | 50 (31.6) | | |
| Hospital | 45 (17.2) | 156 (22.6) | 153 (45.7) | 7 (2.6) | 58 (36.7) | | |
| Other facility | 33 (12.6) | 79 (11.4) | 106 (31.6) | 62 (23.0) | 44 (27.8) | | |
| Other | 3 (1.1) | 25 (3.6) | 22 (6.6) | 5 (1.9) | 6 (3.8) | | |
| **Place of death** | | | | | | | |
| Home | 146 (55.9) | 484 (70.0) | 155 (46.3) | 215 (79.9) | 65 (41.1) | | |
| Hospital | 51 (19.5) | 127 (18.4) | 119 (35.5) | 18 (6.7) | 53 (33.5) | | |
| Other facility | 19 (7.3) | 30 (4.3) | 28 (8.4) | 20 (7.4) | 13 (8.2) | | |
| Other | 45 (17.2) | 49 (7.1) | 33 (9.9) | 15 (5.6) | 27 (17.1) | | |
| Don't know | 0 (0) | 1 (0.1) | 0 (0) | 1 (0.4) | 0 (0) | | |
| **Mother's age** | | | | | | | |
| 10-Dec | 2 (0.8) | 1 (0.1) | 0 (0) | 0 (0) | 0 (0) | | |
| 13-19 | 85 (32.6) | 103 (14.9) | 54 (16.1) | 37 (13.8) | 28 (17.7) | | |
| 20-34 | 138 (52.9) | 463 (67.0) | 217 (64.8) | 174 (64.7) | 90 (57.0) | | |
| 35+ | 29 (11.2) | 120 (17.4) | 48 (14.3) | 37 (13.8) | 37 (23.4) | | |
| Missing | 7 (2.7) | 4 (0.6) | 16 (4.8) | 21 (7.8) | 3 (1.9) | | |
| μ (SE) years | 23.9 (0.5) | 27.1 (0.3) | 26.8 (0.4) | 26.9 (0.5) | 27.6 (0.6) | | |
| **Mother's education** | | | | | | | |
| None | 4 (1.5) | 337 (48.8) | 76 (22.7) | 233 (86.6) | 30 (19.0) | | |
| Primary | 190 (72.8) | 164 (23.7) | 184 (54.9) | 23 (8.6) | 24 (15.2) | | |
| Secondary | 63 (24.1) | 129 (18.7) | 72 (21.5) | 7 (2.6) | 104 (65.8) | | |
| Higher | 0 (0) | 40 (5.8) | 2 (0.6) | 0 (0) | 0 (0) | | |
| Missing | 4 (1.5) | 21 (3.0) | 1 (0.3) | 6 (2.2) | 0 (0) | | |
| **HH* electricity** | | | | | | | |
| Yes | 69 (26.4) | 305 (44.1) | 3 (0.9) | 24 (8.9) | 26 (16.5) | | |
| No | 192 (73.6) | 386 (55.9) | 332 (99.1) | 245 (91.1) | 132 (83.5) | | |
| **HH* floor material** | | | | | | | |
| Mud/clay | 218 (83.5) | 241 (34.9) | 295 (88.1) | 244 (90.7) | 108 (68.4) | | |
| W/C/T** | 43 (16.5) | 422 (61.1) | 40 (11.9) | 25 (9.3) | 50 (31.6) | | |
| Other | 0 (0) | 28 (4.1) | 0 (0) | 0 (0) | 0 (0) | | |

*Household, **Wood/cement/tiles

PLOS Global Public Health

as these two countries had the lowest proportions of teenage mothers except for mothers of neonates in Pakistan. Table 5 shows mothers' mean ages and years of schooling stratified by formal careseeking for their child's illness.

The high percentages of households in all countries but Pakistan without electricity and with mud or clay floors suggests the extreme poverty of the study households. However, Nigeria, with the second highest level of home deaths, had the highest level of electricity and lowest of mud or clay floors among all the African countries, while also having the second highest level of no maternal education.

### Illness severity classification

Tables 2 and 3 show the association between the 2-sign classification of illness severity at onset and the IMCI and extended IMCI-VASA classifications on illness day-1 for neonatal and 1–11-month-old deaths, respectively. It can be seen in Table 2 that both the IMCI and IMCI-VASA methods classified 82% to 96% of fatal neonatal illnesses as severe on illness day-1; whereas the 2-sign method classified 38% to 61% of all illnesses as severe, 19% to 53% as moderate, and 9% to 24% as mildly ill. Weighted Cohen's Kappa statistic confirmed very low agreement between the 2-sign and corresponding IMCI and IMCI-VASA categories, with Kappas of 0.42 and 0.16 for Tanzania and ranging from 0.01 to 0.31 for the other four countries (Table B in S1 Appendix). Also, among neonates with a severe illness by the IMCI and IMCI-VASA

**Table 2. Association of VASA 2-sign illness severity score with IMCI illness signs requiring treatment and urgent referral and IMCI illness signs plus additional VASA illness signs, and the mean number of IMCI and IMCI-VASA illness signs that led to urgent referral and severe illness assessments, for neonates whose fatal illness started in the community.**

| Country | VASA 2-sign severity at illness onset | IMCI day-1 illness severity | | | | IMCI-VASA day-1 illness signs | | | | Total N (%)* |
|---|---|---|---|---|---|---|---|---|---|---|
| | | Mild illness N (%)* | Moderate illness N (%)* | Urgent referral | | Mild illness N (%)* | Moderate illness N (%)* | Severe illness | | |
| | | | | N (%)* | μ # of signs | | | N (%)* | μ # of signs | |
| Cameroon | Mild | 6 (100) | 1 (10) | 11 (8.9) | 1.364 | 0 | 2 (40.0) | 16 (11.9) | 1.813 | 18 (13.0) |
| | Moderate | 0 | 9 (90) | 27 (22.0) | 2.111 | 0 | 3 (60.0) | 33 (24.6) | 3.121 | 36 (25.9) |
| | Severe | 0 | 0 | 85 (69.1) | 3.306 | 0 | 0 | 85 (63.4) | 5.012 | 85 (61.1) |
| | Total** | 6 (4.3) | 10 (7.2) | 123 (88.5) | | 0 | 5 (3.6) | 134 (96.4) | | 139 (100) |
| Nigeria | Mild | 29 (100) | 3 (4.7) | 17 (3.7) | 1.455 | 23 (100) | 6 (13.1) | 20 (4.1) | 1.592 | 48 (8.8) |
| | Moderate | 0 | 57 (95.3) | 232 (50.6) | 2.378 | 0 | 39 (86.9) | 250 (52.2) | 2.923 | 289 (52.9) |
| | Severe | 0 | 0 | 209 (45.7) | 3.451 | 0 | 0 | 209 (43.6) | 5.015 | 209 (38.3) |
| | Total** | 29 (5.3) | 60 (11.0) | 457 (83.7) | | 23 (4.2) | 44 (8.1) | 479 (87.7) | | 546 (100) |
| Malawi | Mild | 11 (100) | 2 (9.2) | 29 (17.6) | 1.647 | 5 (100) | 2 (12.5) | 36 (19.8) | 1.668 | 43 (21.0) |
| | Moderate | 0 | 24 (90.8) | 56 (33.4) | 2.046 | 0 | 14 (87.5) | 65 (35.5) | 2.492 | 79 (38.9) |
| | Severe | 0 | 0 | 82 (49.1) | 3.403 | 0 | 0 | 82 (44.7) | 3.403 | 82 (40.1) |
| | Total** | 11 (5.4) | 26 (12.8) | 167 (81.7) | | 5 (2.2) | 16 (8.0) | 183 (89.7) | | 204 (100) |
| Niger | Mild | 21 (100) | 2 (9.9) | 39 (10.3) | 2.264 | 6 (100) | 4 (26.1) | 51 (13.0) | 2.877 | 61 (14.7) |
| | Moderate | 0 | 14 (90.1) | 109 (28.7) | 2.618 | 0 | 10 (73.9) | 113 (28.5) | 3.567 | 123 (29.6) |
| | Severe | 0 | 0 | 232 (61.0) | 3.340 | 0 | 0 | 232 (58.5) | 4.800 | 232 (55.7) |
| | Total** | 21 (5.0) | 16 (3.8) | 379 (91.3) | | 6 (1.4) | 14 (3.4) | 396 (95.2) | | 416 (100) |
| Tanzania | Mild | 15 (100) | 1 (22.4) | 11 (11.4) | 2.179 | 7 (100) | 2 (38.9) | 18 (17.5) | 2.508 | 26 (23.5) |
| | Moderate | 0 | 3 (77.6) | 18 (19.3) | 2.300 | 0 | 2 (61.1) | 19 (18.8) | 3.496 | 21 (19.1) |
| | Severe | 0 | 0 | 65 (69.3) | 3.571 | 0 | 0 | 65 (63.8) | 5.197 | 65 (57.4) |
| | Total** | 15 (13.1) | 4 (4.0) | 93 (82.9) | | 7 (6.4) | 4 (3.5) | 101 (90.1) | | 112 (100) |

*column percent; **row percent

**Table 3. Association of VASA 2-sign illness severity score with IMCI illness signs requiring treatment and urgent referral and IMCI illness signs plus additional VASA illness signs, and the mean number of IMCI and IMCI-VASA illness signs that led to urgent referral and severe illness assessments, for 1-11-month-olds whose fatal illness started in the community.**

| Country | VASA 2-sign severity at illness onset | IMCI day-1 illness severity | | | | IMCI-VASA day-1 illness signs | | | | Total N (%)* |
|---|---|---|---|---|---|---|---|---|---|---|
| | | Mild illness N (%)* | Moderate illness N (%)* | Urgent referral | | Mild illness N (%)* | Moderate illness N (%)* | Severe illness | | |
| | | | | N (%)* | μ # of signs | | | N (%)* | μ # of signs | |
| Cameroon | Mild | 81 (94.2) | 21 (19.1) | 6 (9.2) | 1.667 | 48 (94.1) | 34 (32.4) | 26 (24.8) | 1.192 | 108 (41.4) |
| | Moderate | 5 (5.8) | 85 (77.3) | 14 (21.5) | 1.286 | 3 (5.9) | 67 (63.8) | 34 (32.4) | 1.382 | 104 (39.9) |
| | Severe | 0 | 4 (3.6) | 45 (69.2) | 1.422 | 0 | 4 (3.8) | 45 (42.9) | 1.614 | 49 (18.8) |
| | Total** | 86 (33.0) | 110 (42.1) | 65 (24.9) | | 51 (19.5) | 105 (40.2) | 105 (40.2) | | 261 (100) |
| Nigeria | Mild | 84 (98.1) | 11 (2.3) | 5 (4.7) | 1.414 | 65 (97.6) | 18 (4.0) | 18 (10.0) | 1.295 | 100 (14.6) |
| | Moderate | 2 (1.9) | 469 (95.4) | 39 (34.0) | 1.143 | 2 (2.4) | 423 (94.6) | 84 (47.9) | 1.301 | 509 (73.8) |
| | Severe | 0 | 11 (2.2) | 70 (61.2) | 1.232 | 0 | 6 (1.4) | 74 (42.1) | 1.477 | 80 (11.7) |
| | Total** | 85 (12.3) | 491 (71.2) | 114 (16.5) | | 67 (9.7) | 447 (64.8) | 176 (25.5) | | 690β (100) |
| Malawi | Mild | 90 (97.9) | 19 (13.0) | 24 (23.8) | 1.205 | 63 (98.8) | 38 (28.4) | 32 (23.1) | 1.662 | 132 (39.5) |
| | Moderate | 2 (2.1) | 120 (83.6) | 19 (19.1) | 1.249 | 1 (1.2) | 94 (70.3) | 46 (33.2) | 1.392 | 141 (42.0) |
| | Severe | 0 | 5 (3.4) | 57 (57.0) | 1.449 | 0 | 2 (1.3) | 60 (43.7) | 1.986 | 62 (17.9) |
| | Total** | 92 (27.5) | 143 (42.8) | 100 (29.8) | | 63 (18.9) | 134 (40.1) | 137 (41.0) | | 335 (100) |
| Niger | Mild | 36 (93.2) | 16 (12.6) | 21 (20.7) | 1.107 | 21 (91.2) | 20 (17.1) | 33 (25.3) | 1.279 | 74 (27.4) |
| | Moderate | 3 (6.8) | 100 (79.2) | 13 (12.5) | 1.092 | 2 (8.8) | 87 (74.0) | 27 (20.8) | 1.138 | 115 (42.9) |
| | Severe | 0 | 10 (8.2) | 69 (66.9) | 1.425 | 0 | 10 (8.9) | 69 (53.9) | 1.727 | 80 (29.7) |
| | Total** | 39 (14.5) | 126 (46.9) | 104 (38.6) | | 23 (8.6) | 117 (43.5) | 129 (47.9) | | 269 (100) |
| Tanzania | Mild | 51 (98.2) | 10 (16.1) | 12 (30.0) | 1.100 | 28 (96.9) | 15 (27.3) | 30 (40.5) | 1.520 | 73 (46.6) |
| | Moderate | 1 (1.8) | 52 (80.3) | 5 (12.4) | 1.663 | 1 (3.1) | 35 (68.1) | 22 (28.8) | 1.407 | 58 (36.9) |
| | Severe | 0 | 2 (3.7) | 24 (57.6) | 1.544 | 0 | 2 (4.6) | 24 (30.7) | 2.249 | 26 (16.5) |
| | Total** | 52 (32.8) | 65 (41.2) | 41 (26.0) | | 29 (18.5) | 53 (33.4) | 76 (48.1) | | 157β (100) |

*column percent; **row percent; β1 missing due to missing values for feeding behavior and activity level

methods, Table 2 reveals a uniformly upward trend in the mean number, respectively, of IMCI urgent referral and IMCI-VASA severe illness signs for neonates with an illness classified by the 2-sign method as mild, moderate, and severe.

The sensitivity test of the 2-sign method was conducted among the 45%, 33%, 32%, 25%, and 54% of all neonates who died on day-0 or day-1 of life, respectively, in Cameroon, Nigeria, Malawi, Niger, and Tanzania. While 61%, 38%, 40%, 56%, and 57% of all neonates in these countries were classified by the 2-sign method as being severely ill at onset (Table 2), 84%, 71%, 69%, 75%, and 81% of neonates that died on day-0 or day-1 were classified as having a severe illness.

Table 3 shows that the IMCI and IMCI-VASA illness signs classified more illnesses of 1–11-month-olds on illness day-1 as mild (9% to 33%) or moderate (33% to 71%) than they did for neonatal illnesses (Table 2). Other than for Tanzania, with Kappas of 0.29 and 0.37, there was moderate to strong agreement of from 0.43 to 0.78 between the 2-sign and corresponding IMCI and IMCI-VASA categories (Table B in S1 Appendix). The 2-sign method also did not as clearly or uniformly separate IMCI and IMCI-VASA severe illnesses characterized by fewer and more signs of IMCI urgent referral and IMCI-VASA severe illness as it did with neonatal illnesses.

We examined caregivers' response to recognition of neonates' moderate and severe illnesses based on the 2-sign method of classifying illness severity, and the response to 1–11-month-old infants' illnesses based on the IMCI-VASA classification method. The IMCI-VASA method was selected over the IMCI alone method because it includes additional signs whose presence indicate a moderate or severe illness, and it identified more illnesses as severe at onset in all five

countries. Due to the above-noted inability of the VASA questionnaire utilized in Mozambique and Pakistan to implement the IMCI-VASA classification method, caregivers' response to illness severity in those two countries was examined only for illnesses of neonates.

**Utility of the illness-severity score for examining careseeking**

Table 4 shows a significant negative association between illness severity and formal careseeking for neonates with a fatal illness in two of the six African countries, and an apparent negative or near-negative association in three other countries

**Table 4. Association of perceived severity at illness onset or illness day-1 with age at death and formal careseeking for fatal illnesses, respectively, of neonates and 1-11-month-olds that started in the community[Ω].**

| Country | VASA 2-sign severity at illness onset | Total N (%)* | Age at death μ days (SE) | F[ϵ] P-value | Did not seek formal care[£] N (%)** | Sought formal care[£] N (%)** | $X^2$ P-value | IMCI-VASA day-1 illness signs | Total N (%)* | Age at death μ months (SE) | F[ϵ] P-value | Did not seek formal care[£] N (%)** | Sought formal care[£] N (%)** | $X^2$ P-value |
|---|---|---|---|---|---|---|---|---|---|---|---|---|---|---|
| **Cameroon** | Mild | 18 (13.0) | 8.6 (1.2) | 18.40 <0.001 | 6 (33.3) | 12 (66.7) | 7.76[β] 0.021 | Mild | 51 (19.5) | 6.5 (0.4) | 0.28 0.754 | 12 (23.5) | 39 (76.5) | 4.63[β] 0.099 |
| | Moderate | 36 (25.9) | 7.3 (1.1) | | 19 (52.8) | 17 (47.2) | | Moderate | 105 (40.2) | 6.7 (0.3) | | 17 (16.2) | 88 (83.8) | |
| | Severe | 85 (61.1) | 2.6 (0.5) | | 57 (67.1) | 28 (32.9) | | Severe | 105 (40.2) | 6.4 (0.3) | | 30 (28.6) | 75 (71.4) | |
| | Total | 139 (100) | 4.6 (0.5) | | 82 (59.0) | 57 (41.0) | | Total | 261 (100) | 6.5 (0.2) | | 59 (22.6) | 202 (77.4) | |
| **Nigeria** | Mild | 48 (8.8) | 8.1 (0.8) | 62.51 <0.001 | 41 (85.0) | 7 (15.0) | 43.27 <0.001 | Mild | 68 (9.8) | 5.6 (0.5) | 2.70 0.068 | 37 (54.6) | 31 (45.4) | 9.50 0.009 |
| | Moderate | 289 (52.9) | 7.2 (0.3) | | 175 (60.6) | 114 (39.4) | | Moderate | 447 (64.7) | 6.1 (0.2) | | 157 (35.1) | 290 (64.9) | |
| | Severe | 209 (38.3) | 2.1 (0.3) | | 184 (87.9) | 25 (12.1) | | Severe | 176 (25.5) | 5.0 (0.3) | | 78 (44.2) | 98 (55.8) | |
| | Total | 546 (100) | 5.3 (0.3) | | 400 (73.2) | 146 (26.8) | | Total | 691 (100) | 5.7 (0.1) | | 272 (39.4) | 419 (60.6) | |
| **Malawi** | Mild | 43 (21.0) | 11.9 (1.3) | 15.61 <0.001 | 16 (37.5) | 27 (62.5) | 6.31 0.043 | Mild | 63 (18.9) | 5.4 (0.4) | 1.30 0.275 | 5 (8.4) | 58 (91.6) | 4.02 0.134 |
| | Moderate | 79 (38.9) | 7.4 (0.9) | | 33 (42.0) | 46 (58.0) | | Moderate | 134 (40.1) | 6.0 (0.3) | | 13 (9.4) | 122 (90.6) | |
| | Severe | 82 (40.1) | 4.3 (0.8) | | 48 (58.2) | 34 (41.8) | | Severe | 137 (41.0) | 5.8 (0.3) | | 23 (16.7) | 114 (83.3) | |
| | Total | 204 (100) | 7.1 (0.6) | | 97 (47.6) | 107 (52.4) | | Total | 335 (100) | 5.8 (0.2) | | 41 (12.2) | 294 (87.8) | |
| **Niger** | Mild | 61 (14.7) | 8.2 (0.9) | 17.74 <0.001 | 41 (66.8) | 20 (33.2) | 4.21 0.122 | Mild | 23 (8.6) | 5.3 (0.7) | 0.46 0.634 | 10 (41.6) | 13 (58.4) | 2.62 0.270 |
| | Moderate | 123 (29.6) | 9.5 (0.9) | | 67 (54 6) | 56 (45.4) | | Moderate | 117 (43.5) | 5.9 (0.3) | | 27 (23.3) | 90 (76.7) | |
| | Severe | 232 (55.7) | 5.2 (0.5) | | 154 (66.3) | 78 (33.7) | | Severe | 129 (47.9) | 5.2 (0.3) | | 40 (30.8) | 89 (69.2) | |
| | Total | 416 (100) | 6.9 (0.4) | | 262 (62.9) | 154 (37.1) | | Total | 269 (100) | 5.5 (0.2) | | 77 (28.5) | 192 (71.5) | |
| **Tanzania** | Mild | 26 (23.5) | 9.0 (1.6) | 11.51 <0.001 | 15 (56.5) | 11 (43.5) | 5.76 0.056 | Mild | 30 (18.8) | 5.2 (0.7) | 2.23 0.111 | 3 (8.6) | 27 (91.4) | 5.62 0.060 |
| | Moderate | 21 (19.1) | 4.5 (1.0) | | 7 (31.5) | 15 (68.5) | | Moderate | 53 (33.3) | 6.5 (0.4) | | 5 (9.9) | 47 (90.1) | |
| | Severe | 65 (57.4) | 1.8 (0.4) | | 43 (66.3) | 22 (33.7) | | Severe | 76 (47.9) | 5.3 (0.5) | | 17 (22.2) | 59 (77.8) | |
| | Total | 112 (100) | 4.0 (0.6) | | 64 (57.4) | 48 (42.6) | | Total | 158 (100) | 5.7 (0.3) | | 25 (15.5) | 133 (84.5) | |
| **Mozambique** | Mild | 105 (39.4) | 7.8 (0.8) | 7.67 0.001 | 85 (81.0) | 20 (19.0) | 11.64 0.003 | | | | | | | |
| | Moderate | 66 (24.7) | 7.2 (0.8) | | 46 (69.5) | 20 (30.5) | | | | | | | | |
| | Severe | 95 (35.8) | 3.7 (0.6) | | 87 (91.6) | 8 (8.4) | | | | | | | | |
| | Total | 265 (100) | 6.2 (0.5) | | 217 (82.0) | 48 (18.0) | | | | | | | | |
| **Pakistan** | Mild | 267 (21.8) | 7.1 (0.4) | 23.86 <0.001 | 162 (60.7) | 105 (39.3) | 14.42 <0.001 | | | | | | | |
| | Moderate | 298 (24.3) | 6.3 (0.4) | | 151 (50.7) | 147 (49.3) | | | | | | | | |
| | Severe | 662 (54.0) | 4.2 (0.2) | | 312 (47.1) | 350 (52.9) | | | | | | | | |
| | Total | 1227 (100) | 5.4 (0.2) | | 625 (50.9)) | 602 (49.1) | | | | | | | | |

[Ω] 25, 177, 116, 37, 116, 137, and 861 neonatal deaths started in the delivery hospital, respectively, in Cameroon, Nigeria, Malawi, Niger, Tanzania, Mozambique, and Pakistan; Total Ns for neonates also exclude cases missing data for illness severity or formal careseeking (see Table C in S1 Appendix); *column percent; [ϵ]Anova; [£]Did not seek formal care at any time during the illness, and Sought formal care during the illness; **row percent; [β]Pearson chi-square (all other chi-squares are Rao-Scott)

that is punctuated by greater careseeking for moderate than either mild or severe illness (Fig 1a). There was also a negative association between age and illness severity in these five countries. In Pakistan the opposite relationship was found between formal careseeking and illness severity—there was a significant positive association; while, just as in the African countries, illness severity increased with decreasing age at death (Table 4 and Fig 2a).

The association of illness severity with careseeking was less definitive for 1–11-month-olds than for neonates. Only in Tanzania was there a (nearly) significant negative association between illness severity and formal careseeking; while in Nigeria, like the finding for neonates, careseeking was greater for moderately ill than either mildly or severely ill children. This was also true in Cameroon and Niger but not significantly so (Table 4 and Fig 1b). Also, contrary to the finding for neonates, there was no significant association between age and illness severity, though moderately ill children were uniformly somewhat older than either the mildly or severely ill (Table 4 and Fig 2b).

Based on the < 10%/ ≥ 10% missing data points criteria, Table 5 presents the results of multivariable logistic regression IM analyses for Mozambique and Pakistan and CC analyses for all other countries. Tables C-F in S1 Appendix present the percentage of missing values for all variables, and Table G in S1 Appendix provides the CC analyses for Mozambique and Pakistan and IM analyses for all other countries. The Table 5 results are favored due to the low level of missing data in all countries but Mozambique and Pakistan.

Moderate illness severity was associated with increased careseeking for neonates in Nigeria, Tanzania, Mozambique and Pakistan; but only in Pakistan was severe illness an apparent driver of careseeking. Older age was positively associated with careseeking in all seven countries, ranging from an increase of 6% in Niger to 11% in Cameroon for each additional day. For 1–11-month-olds, moderate illness severity doubled formal careseeking in Nigeria; while older age increased careseeking in three countries, by from 10% in Nigeria to 26% in Tanzania for each additional month, and nearly significantly so in Niger, by 12% (Fig 2b).

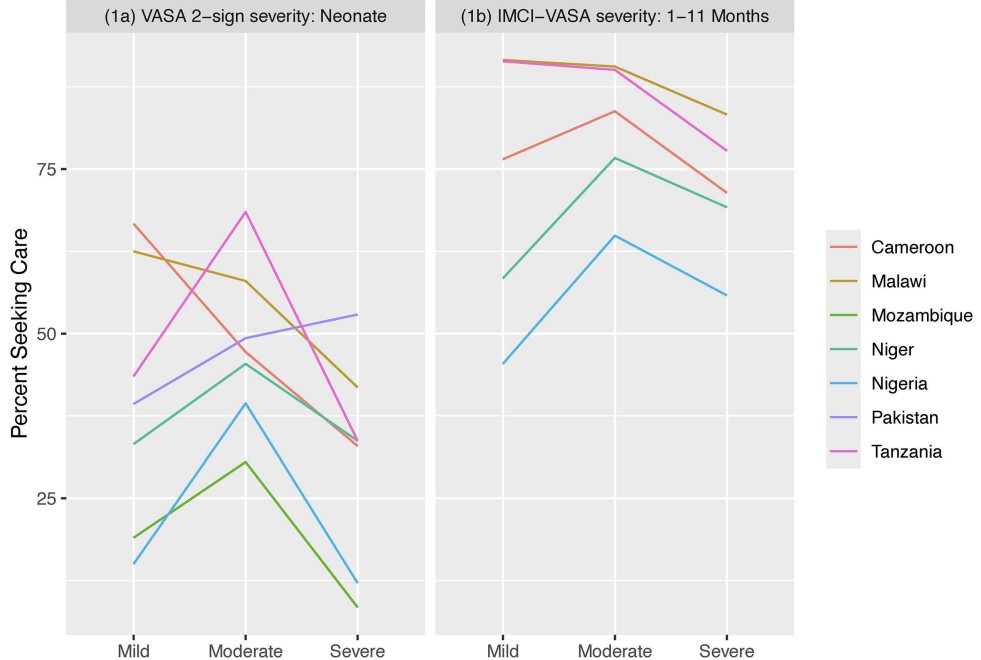

**Fig 1. Percent of neonates (1a) and 1–11-month-olds (1b) with a mild, moderate and severe illness at onset or on illness day-1, respectively, by the VASA 2-sign and IMCI-VASA illness severity methods, who sought formal health care.**

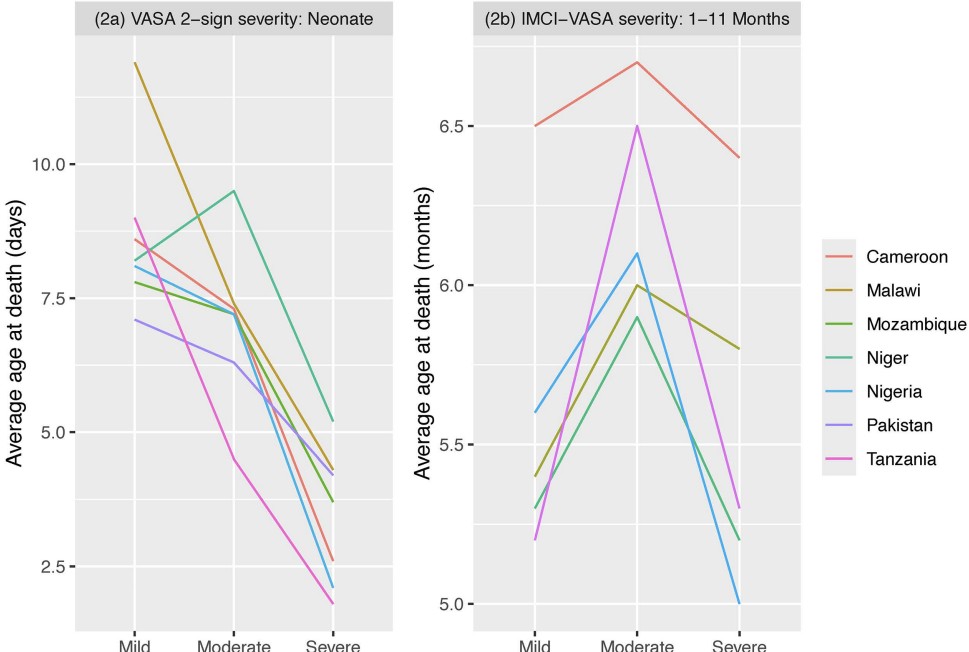

**Fig 2. The average age at death of neonates (2a) and 1–11-month-olds (2b) with a mild, moderate and severe illness at onset or on illness day-1, respectively, by the VASA 2-sign and IMCI-VASA illness severity methods.**

Cause of death played a significant role only in Nigeria and in Malawi, where neonates and 1–11-month-olds with an infectious cause were more than twice as likely to seek formal care. There was also a universal lack of increased careseeking for neonates with IPRE or prematurity, and nearly significant decrease in Niger. Mothers completing more years of school increased careseeking for neonates in four countries and for infants in three countries, by 8% to 59% for each additional year. Each additional hour of travel time to the usual facility used in an emergency decreased formal careseeking by about one-third for infants in two countries. Mother's age significantly increased careseeking just for 1–11-month-olds in Nigeria, by 3% for each added year.

Going by the Table G in S1 Appendix findings would exclude Pakistan from the countries where careseeking was increased for older neonatal age, thus decreasing this effect from all seven countries to six, while adding Niger as a fourth country where older infant age boosted careseeking. This would also add Pakistan as a third country where careseeking was increased for infants' severe infections and move Niger from a near decrease to a decrease in careseeking for IPRE or prematurity. Finally, using Table G's (in S1 Appendix) findings would delete the positive effect of older maternal age on careseeking for infants in Nigeria.

## Discussion

We developed a simple 2-sign score capable of identifying mild, moderate, or severe neonatal illness based on feeding behavior and activity level, signs that have previously been shown to be both objective indicators of neonatal illness severity [7–9] and easily recognized motivators of careseeking for sick young children [5,6]. Caregivers have also been found to recognize inability to play, perhaps correlating with our 'less active' or 'not moving' signs, as indicators of severe fever in children under 5 years in Malawi [27]. Similarly, studies in India and Ghana have found lethargy to be one of the few signs of neonatal illness both easily recognized and understood by mothers to indicate the need for formal health care [28,29]. While the extreme levels of these signs are included among the IMCI illness signs requiring urgent referral, and

**Table 5. Multivariable logistic regression of factors associated with seeking formal health care for neonates and 1-11-month-olds with a fatal illness[Ω].**

| Country<br>Explanatory factors | Neonates (0–27 days) | | | | 1-11-month-olds | | | |
|---|---|---|---|---|---|---|---|---|
| | Did not seek formal care[£]<br>N (%) | Sought formal care[£]<br>N (%) | p-value* | AOR (95% CI) | Did not seek formal care[£]<br>N (%) | Sought formal care[£]<br>N (%) | p-value* | AOR (95% CI) |
| **Cameroon** | 79 (59.4) | 54 (40.6) | | | 54 (22.1) | 190 (77.9) | | |
| Neonatal cause of death | | | | | | | | |
| All other causes | 11 (13.9) | 5 (9.3) | -- | 1.0 (ref[β]) | -- | -- | -- | -- |
| IPRE[€] or prematurity | 46 (58.2) | 22 (40.7) | 0.731 | 1.26 (0.34, 4.61) | -- | -- | -- | -- |
| Severe infection | 22 (27.9) | 27 (50.0) | 0.134 | 2.69 (0.74, 9.84) | -- | -- | -- | -- |
| Infant cause of death | | | | | | | | |
| All other causes | -- | -- | -- | -- | 21 (38.9) | 70 (36.8) | -- | 1.0 (ref) |
| Severe febrile infection | -- | -- | -- | -- | 33 (61.1) | 120 (63.2) | 0.698 | 1.14 (0.60, 2.15) |
| Neonatal age at death<br>Mean days (SE**) | 3.04 (0.47) | 6.37 (0.86) | 0.017 | 1.11 (1.02, 1.21) | -- | -- | -- | -- |
| Infant age at death<br>Mean months (SE) | -- | -- | -- | -- | 5.59 (0.43) | 6.77 (0.22) | 0.024 | 1.13 (1.02, 1.25) |
| Illness severity[±] | | | | | | | | |
| Mild | 6 (7.6) | 11 (20.4) | -- | 1.0 (ref) | 12 (22.2) | 38 (20.0) | -- | 1.0 (ref) |
| Moderate | 19 (24.1) | 17 (31.5) | 0.288 | 0.50 (0.14, 1.80) | 15 (27.8) | 80 (42.1) | 0.319 | 1.55 (0.66, 3.67) |
| Severe | 54 (68.4) | 26 (48.2) | 0.154 | 0.40 (0.11, 1.41) | 27 (50.0) | 72 (37.9) | 0.667 | 0.84 (0.37, 1.88) |
| Travel to usual facility<br>Mean hours (SE) | 0.70 (0.09) | 0.58 (0.08) | 0.768 | 0.92 (0.54, 1.59) | 0.78 (0.10) | 0.81 (0.06) | 0.486 | 1.15 (0.78, 1.70) |
| Mother's age<br>Mean years (SE) | 22.32 (0.79) | 22.09 (1.13) | 0.959 | 1.00 (0.95, 1.05) | 24.93 (1.05) | 23.72 (0.52) | 0.416 | 0.98 (0.94, 1.02) |
| Mother's education<br>Mean years (SE) | 5.82 (0.23) | 6.07 (0.27) | 0.036 | 1.25 (1.02, 1.54) | 5.17 (0.26) | 5.41 (0.15) | 0.773 | 1.02 (0.87, 1.20) |
| **Nigeria** | 381 (72.6) | 144 (27.4) | | | 254 (38.6) | 404 (61.4) | | |
| Neonatal cause of death | | | | | | | | |
| All other causes | 101 (26.4) | 30 (20.9) | -- | 1.0 (ref) | -- | -- | -- | -- |
| IPRE or prematurity | 107 (28.0) | 18 (12.7) | 0.827 | 0.91 (0.38, 2.17) | -- | -- | -- | -- |
| Severe infection | 174 (45.6) | 95 (66.4) | 0.086 | 1.66 (0.93, 2.96) | -- | -- | -- | -- |
| Infant cause of death | | | | | | | | |
| All other causes | -- | -- | -- | -- | 86 (33.9) | 118 (29.1) | -- | 1.0 (ref) |
| Severe febrile infection | -- | -- | -- | -- | 168 (66.1) | 287 (70.9) | 0.019 | 1.60 (1.08, 2.36) |
| Neonatal age at death<br>Mean days (SE) | 4.28 (0.28) | 7.60 (0.52) | 0.001 | 1.07 (1.03, 1.11) | -- | -- | -- | -- |
| Infant age at death<br>Mean months (SE) | -- | -- | -- | -- | 5.20 (0.20) | 5.94 (0.17) | 0.003 | 1.10 (1.03, 1.16) |
| Illness severity[±] | | | | | | | | |
| Mild | 40 (10.5) | 7 (4.6) | -- | 1.0 (ref) | 35 (13.6) | 31 (7.6) | -- | 1.0 (ref) |
| Moderate | 164 (42.9) | 112 (77.8) | 0.005 | 4.35 (1.59, 11.93) | 143 (56.3) | 278 (68.8) | 0.016 | 2.24 (1.17, 4.30) |
| Severe | 177 (46.0) | 25 (17.6) | 0.520 | 1.46 (0.46, 4.60) | 76 (30.1) | 95 (23.6) | 0.200 | 1.57 (0.79, 3.11) |
| Travel to usual facility<br>Mean hours (SE) | 0.89 (0.05) | 0.83 (0.07) | 0.528 | 0.89 (0.61, 1.29) | 0.93 (0.05) | 0.90 (0.17) | 0.672 | 0.97 (0.86, 1.11) |
| Mother's age<br>Mean years (SE) | 25.84 (0.39) | 26.99 (0.67) | 0.231 | 1.02 (0.99, 1.06) | 25.84 (0.49) | 27.62 (0.38) | 0.045 | 1.03 (1.00, 1.05) |
| Mother's education<br>Mean years (SE) | 2.50 (0.22) | 4.03 (0.39) | 0.004 | 1.08 (1.03, 1.14) | 2.18 (0.22) | 5.00 (0.26) | <0.001 | 1.15 (1.10, 1.21) |

*(Continued)*

**Table 5.** (Continued)

| Country Explanatory factors | Neonates (0–27 days) | | | | 1-11-month-olds | | | |
|---|---|---|---|---|---|---|---|---|
| | Did not seek formal care£ N (%) | Sought formal care£ N (%) | p-value* | AOR (95% CI) | Did not seek formal care£ N (%) | Sought formal care£ N (%) | p-value* | AOR (95% CI) |
| **Malawi** | 93 (46.8) | 105 (53.2) | | | 35 (11.1) | 280 (88.9) | | |
| Neonatal cause of death | | | | | | | | |
| All other causes | 31 (33.5) | 19 (17.8) | -- | 1.0 (ref) | -- | -- | -- | -- |
| IPRE or prematurity | 29 (31.0) | 24 (22.4) | 0.105 | 2.20 (0.85, 5.72) | -- | -- | -- | -- |
| Severe infection | 33 (35.5) | 63 (59.8) | 0.017 | 2.73 (1.20, 6.18) | -- | -- | -- | -- |
| Infant cause of death | | | | | | | | |
| All other causes | -- | -- | -- | -- | 19 (54.7) | 95 (34.0) | -- | 1.0 (ref) |
| Severe febrile infection | -- | -- | -- | -- | 16 (45.3) | 184 (66.0) | 0.014 | 2.73 (1.23, 6.06) |
| Neonatal age at death Mean days (SE) | 4.39 (0.64) | 9.47 (0.89) | 0.001 | 1.10 (1.04, 1.16) | -- | -- | -- | -- |
| Infant age at death Mean months (SE) | -- | -- | -- | -- | 5.04 (0.57) | 5.92 (0.20) | 0.128 | 1.11 (0.97, 1.27) |
| Illness severity± | | | | | | | | |
| Mild | 16 (17.4) | 27 (25.4) | 0.460 | 1.0 (ref) | 4 (12.8) | 55 (19.7) | -- | 1.0 (ref) |
| Moderate | 33 (35.3) | 45 (42.7) | 0.771 | 1.42 (0.56, 3.57) | 12 (33.9) | 113 (40.5) | 0.911 | 1.08 (0.30, 3.81) |
| Severe | 44 (47.3) | 34 (31.8) | | 0.87 (0.34, 2.24) | 19 (53.3) | 111 (39.8) | 0.426 | 0.60 (0.17, 2.10) |
| Travel to usual facility Mean hours (SE) | 1.97 (0.16) | 1.84 (0.14) | 0.832 | 1.03 (0.81, 1.29) | 2.23 (0.25) | 1.59 (0.07) | 0.001 | 0.63 (0.48, 0.83) |
| Mother's age Mean years (SE) | 24.96 (0.82) | 24.72 (0.70) | 0.879 | 1.00 (0.95, 1.06) | 25.89 (1.16) | 26.67 (0.46) | 0.432 | 1.02 (0.97, 1.07) |
| Mother's education Mean years (SE) | 3.84 (0.32) | 4.60 (0.33) | 0.085 | 1.10 (0.99, 1.21) | 2.74 (0.45) | 4.14 (0.23) | 0.011 | 1.17 (1.04, 1.33) |
| **Niger** | 238 (62.0) | 145 (38.0) | | | 69 (28.2) | 176 (71.8) | | |
| Neonatal cause of death | | | | | | | | |
| All other causes | 51 (21.3) | 37 (25.5) | -- | 1.0 (ref) | -- | -- | -- | -- |
| IPRE or prematurity | 76 (31.9) | 23 (15.7) | 0.051 | 0.43 (0.18, 1.01) | -- | -- | -- | -- |
| Severe infection | 111 (46.7) | 85 (58.7) | 0.542 | 0.83 (0.46, 1.51) | -- | -- | -- | -- |
| Infant cause of death | | | | | | | | |
| All other causes | -- | -- | -- | -- | 19 (28.1) | 55 (31.2) | -- | 1.0 (ref) |
| Severe febrile infection | -- | -- | -- | -- | 50 (71.9) | 121 (68.8) | 0.708 | 0.89 (0.48, 1.65) |
| Neonatal age at death Mean days (SE) | 5.80 (0.51) | 9.10 (0.83) | 0.002 | 1.06 (1.02, 1.10) | -- | -- | -- | -- |
| Infant age at death Mean months (SE) | -- | -- | -- | -- | 4.63 (0.47) | 5.75 (0.29) | 0.054 | 1.12 (0.998, 1.26) |
| Illness severity± | | | | | | | | |
| Mild | 38 (15.9) | 19 (13.0) | -- | 1.0 (ref) | 10 (13.9) | 11 (6.5) | -- | 1.0 (ref) |
| Moderate | 61 (25.8) | 51 (35.3) | 0.216 | 1.60 (0.76, 3.38) | 25 (36.5) | 86 (48.7) | 0.136 | 2.52 (0.75, 8.51) |
| Severe | 139 (58.4) | 75 (51.7) | 0.304 | 1.50 (0.69, 3.26) | 34 (49.7) | 79 (44.8) | 0.196 | 1.99 (0.70, 5.66) |
| Travel to usual facility Mean hours (SE) | 1.43 (0.12) | 1.04 (0.17) | 0.270 | 0.83 (0.60, 1.15) | 1.57 (0.19) | 0.84 (0.08) | <0.001 | 0.60 (0.45, 0.81) |
| Mother's age Mean years (SE) | 26.32 (0.51) | 25.62 (0.59) | 0.246 | 0.98 (0.95, 1.01) | 26.82 (0.79) | 27.36 (0.80) | 0.962 | 1.00 (0.96, 1.04) |
| Mother's education Mean years (SE) | 0.46 (0.11) | 0.71 (0.19) | 0.199 | 1.09 (0.96, 1.24) | 0.31 (0.15) | 0.95 (0.35) | 0.0498 | 1.18 (1.00, 1.39) |

*(Continued)*

| Country Explanatory factors | Neonates (0–27 days) | | | | 1-11-month-olds | | | |
|---|---|---|---|---|---|---|---|---|
| | Did not seek formal care£ N (%) | Sought formal care£ N (%) | p-value* | AOR (95% CI) | Did not seek formal care£ N (%) | Sought formal care£ N (%) | p-value* | AOR (95% CI) |
| **Tanzania** | 64 (57.4) | 48 (42.6) | | | 23 (14.6) | 133 (85.4) | | |
| Neonatal cause of death | | | | | | | | |
| All other causes | 22 (33.9) | 10 (21.9) | -- | 1.0 (ref) | -- | -- | -- | -- |
| IPRE or prematurity | 23 (36.2) | 11 (22.7) | 0.627 | 1.51 (0.28, 8.33) | -- | -- | -- | -- |
| Severe infection | 19 (29.9) | 27 (55.4) | 0.086 | 3.39 (0.84, 13.77) | -- | -- | -- | -- |
| Infant cause of death | | | | | | | | |
| All other causes | -- | -- | -- | -- | 10 (45.7) | 43 (32.6) | -- | 1.0 (ref) |
| Severe febrile infection | -- | -- | -- | -- | 12 (54.3) | 89 (67.4) | 0.294 | 1.80 (0.60, 5.45) |
| Neonatal age at death Mean days (SE) | 2.89 (0.67) | 5.54 (0.99) | 0.043 | 1.09 (1.003, 1.19) | -- | -- | -- | -- |
| Infant age at death Mean months (SE) | -- | -- | -- | -- | 3.84 (0.70) | 5.99 (0.34) | 0.046 | 1.26 (1.01, 1.58) |
| Illness severity± | | | | | | | | |
| Mild | 15 (23.2) | 11 (24.0) | -- | 1.0 (ref) | 3 (11.4) | 27 (20.5) | -- | 1.0 (ref) |
| Moderate | 7 (10.5) | 15 (30.6) | 0.042 | 4.83 (1.06, 21.96) | 5 (23.0) | 47 (35.1) | 0.561 | 0.66 (0.16, 2.76) |
| Severe | 43 (66.4) | 22 (45.4) | 0.595 | 1.45 (0.36, 5.87) | 15 (65.6) | 59 (44.4) | 0.186 | 0.40 (0.10, 1.58) |
| Travel to usual facility Mean hours (SE) | 0.84 (0.14) | 1.57 (0.79) | 0.077 | 1.15 (0.98, 1.35) | 0.75 (0.15) | 0.78 (0.07) | 0.895 | 0.95 (0.42, 2.13) |
| Mother's age Mean years (SE) | 27.12 (2.53) | 25.28 (1.14) | 0.632 | 0.99 (0.93, 1.05) | 25.92 (2.06) | 27.72 (0.86) | 0.563 | 1.03 (0.94, 1.12) |
| Mother's education Mean years (SE) | 5.74 (0.62) | 6.39 (0.43) | 0.101 | 1.14 (0.97, 1.33) | 4.69 (0.73) | 5.82 (0.28) | 0.284 | 1.07 (0.95, 1.20) |
| **Mozambique** | 217 (82.0) | 48 (18.0) | | | | | | |
| Neonatal cause of death | | | | | | | | |
| All other causes | 64 (29.6) | 9 (18.5) | -- | 1.0 (ref) | | | | |
| IPRE or prematurity | 73 (33.8) | 14 (29.6) | 0.253 | 1.86 (0.64, 5.37) | | | | |
| Severe infection | 80 (36.6) | 25 (51.9) | 0.057 | 2.59 (0.97, 6.89) | | | | |
| Neonatal age at death Mean days (SE) | 5.24 (0.52) | 10.55 (1.08) | <0.001 | 1.10 (1.04, 1.16) | | | | |
| Illness severity± | | | | | | | | |
| Mild | 85 (39.0) | 20 (41.5) | -- | 1.0 (ref) | | | | |
| Moderate | 46 (21.0) | 20 (41.8) | 0.022 | 3.05 (1.18, 7.92) | | | | |
| Severe | 87 (40.1) | 8 (16.7) | 0.290 | 0.57 (0.20, 1.62) | | | | |
| Travel to usual facility Mean hours (SE) | 9.00 (2.57) | 3.41 (0.88) | 0.273 | 0.97 (0.92, 1.03) | | | | |
| Mother's age Mean years (SE) | 25.11 (0.82) | 25.57 (1.59) | 0.416 | 1.02 (0.97, 1.09) | | | | |
| Mother's education Mean years (SE) | 3.79 (0.20) | 5.60 (0.39) | <0.001 | 1.59 (1.32, 1.90) | | | | |

*(Continued)*

**Table 5.** (Continued)

| Country<br>Explanatory factors | Neonates (0–27 days) | | | | 1-11-month-olds | | | |
|---|---|---|---|---|---|---|---|---|
| | Did not seek formal care£<br>N (%) | Sought formal care£<br>N (%) | p-value* | AOR (95% CI) | Did not seek formal care£<br>N (%) | Sought formal care£<br>N (%) | p-value* | AOR (95% CI) |
| **Pakistan** | 625 (50.9) | 602 (49.1) | | | | | | |
| Neonatal cause of death | | | | | | | | |
| All other causes | 233 (37.3) | 195 (32.4) | -- | 1.0 (ref) | | | | |
| IPRE or prematurity | 176 (28.2) | 150 (24.9) | 0.841 | 0.97 (0.71, 1.32) | | | | |
| Severe infection | 216 (34.6) | 257 (42.7) | 0.084 | 1.28 (0.97, 1.69) | | | | |
| Neonatal age at death<br>Mean days (SE) | 4.31 (0.24) | 6.46 (0.28) | <0.001 | 1.07 (1.04, 1.09) | | | | |
| Illness severity± | | | | | | | | |
| Mild | 162 (25.9) | 105 (17.4) | -- | 1.0 (ref) | | | | |
| Moderate | 151 (24.2) | 147 (24.4) | 0.006 | 1.65 (1.16, 2.36) | | | | |
| Severe | 312 (49.9) | 350 (58.1) | <0.001 | 2.22 (1.62, 3.03) | | | | |
| Travel to usual facility<br>Mean hours (SE) | 1.40 (0.12) | 0.97 (0.11) | 0.085 | 0.96 (0.91, 1.01) | | | | |
| Mother's age<br>Mean years (SE) | 26.81 (0.30) | 26.53 (0.23) | 0.241 | 0.99 (0.97, 1.01) | | | | |
| Mother's education<br>Mean years (SE) | 7.03 (0.07) | 7.95 (0.09) | <0.001 | 1.29 (1.18, 1.40) | | | | |

ΩComplete case analyses for Cameroon (missing 4.3% neonatal and 6.5% 1–11-months-old data points), Nigeria (missing 3.8% and 4.8%), Malawi (missing 2.9% and 6.0%), Niger (missing 7.7% and 8.9%), and Tanzania (missing 0% and 1.3%); Analyses with imputed means for travel time, mother's age and/or mother's schooling for Mozambique and Pakistan; Total Ns for neonates also exclude cases missing data for illness severity or formal careseeking (see Table C in S1 Appendix); £Did not seek formal care at any time during the illness, and Sought formal care during the illness; ±Illness severity at onset (neonates) or on illness day-1 (1–11-month-olds); *Anova F-value for continuous variables, X2 for categorical variables; **Standard Error; βReference for each other level; €IPRE: Intrapartum-related event (birth asphyxia or injury)

poor feeding is an IMCI other illness sign, we found that the 2-sign scoring system was better able to differentiate mothers' perception of mild, moderate, and severe neonatal illness at onset or on illness day-1 than were any one or more of 33 IMCI-VASA signs of moderate and severe illness. Our sensitivity test of the 2-sign method reinforced the validity of its assessments of illness severity of neonates by finding, as hypothesized, that a higher percentage of neonates who died on day-0 or day-1 of life were scored as being severely ill at onset. In contrast, one or more of 29 IMCI-VASA signs better distinguished mild, moderate, and severe illness of 1–11-month-olds than did the 2-sign method. This is perhaps to be expected given that illness in neonates is often characterized by less specific signs than in sick older children.

Verbal autopsy studies in general, as well as other studies, have assessed illness severity using individual signs for particular illnesses, for example, more than six loose stools in one day for severe diarrhea [24]. In addition to operating on only one illness at a time, this method also does not identify moderate illnesses more likely to be amenable to effective treatment. In contrast, our 2-sign and IMCI-VASA signs methods provide global measures of moderate and severe illness for all major causes of neonatal and infant mortality in low- and middle-income countries [30].

The 2-sign method identified a range of mild, moderate, and severe neonatal illnesses at onset or on illness day-1 in all seven study countries, demonstrating that caregivers were able to report various levels of these illness signs. This also enabled determining the association of these three levels of illness severity with formal careseeking. Only in Pakistan was

there increased careseeking for both moderate and severe illness. The univariate negative association between careseeking and illness severity in two African countries and apparent negative association in three others was revealed by the multivariable analysis to in fact be a positive association between careseeking and moderate illness in three countries and no association between careseeking and the severity level in two countries.

Some authors have found that careseeking for young children is initiated only once illness reaches such a severe state that there is little chance of health care saving the child [12], and that this is due in part to lack of recognition or failing to act on signs of moderate illness [6]. Others have found decreased careseeking for severely ill neonates due in part to caregivers' concern for their newborn's fragility and the danger of taking such a young, sick child outside the home [31]. This explanation is also implied by our finding that, while neonatal age was uniformly inversely related to illness severity, in multivariable analyses careseeking was increased for older neonates in all seven countries but was increased for severe illness only in Pakistan. This suggests that caregivers in Pakistan were responding to their neonates' severe illness, despite, or perhaps even more so because of, their young age; and warrants further study of the cultural or other factors driving this difference in careseeking behavior.

Increased careseeking for moderately ill neonates in Nigeria, Tanzania, and Mozambique, as well as in Pakistan, was perhaps encouraged by the relatively advanced age of the moderately ill as compared to the severely ill. A related possible explanation warranting further exploration is the perceived greater chance of survival of moderately ill neonates and a sense of fatalism overcoming the drive to seek care for younger, severely ill children. Mothers in a study of fatal childhood illnesses in Indonesia expressed this as "God's will that the child would die" [32]. A counter argument put forth in response to this finding, that fatalism represents a post-hoc response to personally irremediable social constraints [33], leaves this explanation as wanting further analysis. Other possibilities include concern for mothers' condition and cultural prohibitions against traveling soon after giving birth, and longer duration of moderately severe illness providing more time to seek care [32].

Contrary to the finding for neonates, there was no clear association between age and illness severity for 1–11-month-olds, though it appears that moderately ill infants were somewhat older than the mildly and severely ill. Yet, just as for neonates, in multivariable analyses age played the predominant role among 1–11-month-olds, with careseeking increased for older infants in three and nearly so in a fourth of five countries; while it was increased for moderate severity only in Nigeria and neither increased nor decreased for severe illness in all countries. These findings suggest that for infants beyond the neonatal period, another possible factor surpassing the degree of illness severity in the careseeking decision making process is the perceived greater chance of survival for an older child who has already traversed their most vulnerable period.

Our findings for both neonates and infants run counter to those of many studies in low resource settings that have found careseeking was increased for younger children, such as an examination of 258 national surveys datasets [34]. However, all these comparisons were of children 0–23 months versus 24–59 months old with a non-fatal illness and with minimal ability to examine the effect of illness severity. Other studies of careseeking for young children with a non-fatal illness have also found decreased careseeking for older children [35,36]; while a study of fatal illnesses of young children found, like our analysis, that careseeking increased in tandem with age increases [32]. A study of fatal childhood pneumonia deaths in Malawi also found that care was first sought at a health facility more often for children aged 12–59-months than for 1–11-month-olds and older children were more likely to die at a health center or hospital vs infants, who died more often at home [37]. In Taffa's study of non-fatal illnesses [36] 95 percent of caretakers perceived their child's morbidity as of mild to moderate severity, with no distinction between those who sought and did not seek care. The limited assessment of severity by the studies of children's non-fatal illnesses, including secondary analyses of DHS data [38], makes it difficult to compare with studies of fatal illnesses. Our almost uniform findings across seven countries of increased careseeking for older neonates and infants, independent of illness severity, is strong evidence that should be examined for consistency in studies of fatal childhood illness in additional countries.

Regarding the impact of cause of death, similar to our finding of increased careseeking for infectious causes in Nigeria and Malawi, a study of 300 sick children in a slum in Bangladesh found the same in comparison to careseeking for malnourished children [39]. The universal lack of increased careseeking that we found for neonates with IPRE or prematurity, and nearly significant decrease in Niger, could be due to the often-rapid progression of these conditions to a severe state and very young age at which they result in death [40]. Very low levels of careseeking for preterm newborns have been documented in other settings [5,41].

The positive contribution of caregivers' education to formal careseeking, as we found for both age groups in more than half the countries, was also found by two multi-country analyses of acute respiratory infection (ARI) [38] and diarrhea [42], although, again, these were both studies of non-fatal illnesses. Other multi-country studies of childhood diarrhea and ARI [24,43] have determined that health education about illness symptoms, rather than caregivers' general education level, increased appropriate careseeking. While our study was not designed to examine place of death, the apparent association of no maternal education with home death of neonates and 1–11-month-olds seen in Tables 1a and 1b supports our finding of the positive association of maternal education with careseeking. Increased distance [35] and time to reach a health facility [39,44] have previously been found to negatively affect careseeking for young children. Positing that longer expected travel time might discourage careseeking, we included time to reach the usual provider in an emergency in our multivariable models and found this had a negative effect on careseeking only for infants and in just two countries. We could not include delay-1 (time to decide to seek formal health care, having recognized an illness) or delay-2 (time to prepare to travel and to reach formal care, having decided to seek care) in the model because these delays could be assessed only for children for whom care was sought.

## Limitations

Like all VASA studies of child mortality, the data collected and analyzed by this study were derived from interviews of deceased children's caregivers. As such, we could not examine the performance of the severity scoring methods among children who survived an illness. Ideally, this would be done for children with a near-miss illness whose final severity level approximated that of the decedents. Being interview-based, with some interviews conducted beyond four years in five of the study countries, the findings are also subject to the possible influence of faulty recall and biased reporting of events. The severity scores developed by the study were based on caregivers' perception of their children's illness, without the ability to examine the children and objectively evaluate their condition. However, this concern is mitigated by the fact that the signs included in the 2-sign method have been shown to be recognized by and of concern to mothers and to correspond with actual illness severity. The consistent inverse relationship between perceived severity level and neonates' age found by the study also suggests that the method accurately assesses illness severity, although theoretically one could argue that mothers might automatically perceive younger neonates as being more severely ill. The validity of the multiple sign method for 1–11-month-olds also depends on the accuracy of caregivers' reports, so is similarly subject to possible inaccurate perception as well as interview and recall bias. However, the method is based on established indicators of illness severity and contributed to sensible findings, suggesting that it functioned well despite its possible limitations. From a statistical standpoint, power may be limited for some countries when examining the associations of illness severity and other factors with seeking formal health care. Finally, as described in the paper's Methods section, it was not possible to do a comparative analysis of the 2-sign and IMCI-VASA methods' performance in two of the seven study countries.

## Conclusions

The 2-sign and multiple-sign methods developed by this study for assessing caregivers' perception, respectively, of their neonate's and 1–11-month-old's condition clearly differentiated mild, moderate, and severe illness at onset of a fatal illness. Due to the more specific signs associated with illnesses of older infants, the 2-sign method should be used only

for neonates. Including the findings from these methods in multivariable analyses of careseeking showed that caregivers were more likely to seek care for moderately ill, older neonates and infants. Guidance on the illness signs for which care should be sought before a young child's illness progresses to a severe state should be included in child survival messaging and is likely to contribute to decreases in child mortality. The 2-sign method can serve as a practical tool for this purpose for illnesses of neonates, while both the 2-sign and multiple sign methods can be included in mortality and morbidity studies to monitor caregivers' response to their children's illness condition and the impact of careseeking messages on child mortality. Qualitative studies are needed to better understand the reasons for reduced health careseeking for younger neonates and infants with a potentially fatal illness. However, it stands to reason that appropriate delivery and newborn care must be provided at community level to prevent and treat intrapartum-related events such as perinatal asphyxia and other early onset severe neonatal illnesses that progress rapidly and for which caregivers are unable to undertake effective careseeking.

## Supporting information

**S1 Appendix. Table A in S1 Appendix**. Illness that started in the community. Table B in S1 Appendix. Kappa agreement with VASA 2-sign severity at illness onset. Table C in S1 Appendix. Among neonates whose illness started in the community, missing the following variables: Table D in S1 Appendix. Among neonates whose illness started in the community and not missing illness severity or sought formal care, but missing the following variables: Table E in S1 Appendix. Among 1–11-month-olds whose illness started in the community, missing the following variables: Table F in S1 Appendix. Among 1–11-month-olds whose illness started in the community and not missing illness severity or sought formal care, but missing the following variables: Table G in S1 Appendix. Logistic regression of factors associated with seeking formal health care for neonates and 1–11-month-olds with a fatal illness$^{\Omega}$. Table G in S1 Appendix legend. $^{\Omega}$Analyses with imputed means for travel time, mother's age and/or mother's schooling for Cameroon, Nigeria, Malawi, Niger, and Tanzania. Complete case analyses for Mozambique (missing 34.7% cases) and Pakistan (missing 67.9%); Total Ns for neonates also exclude cases missing data for illness severity or formal careseeking (see Table C in S1 Appendix); *Standard Error; $^{\beta}$Reference for each other level; $^{€}$IPRE: Intrapartum-related event (birth asphyxia or birth injury).
(DOCX)

**S1 Data. Nigeria minimal neonatal dataset.**
(XLSX)

**S2 Data. Nigeria minimal 1–11-month-olds dataset.**
(XLSX)

**S3 Data. Malawi minimal neonatal dataset.**
(XLSX)

**S4 Data. Malawi minimal 1–11-month-olds dataset.**
(XLSX)

**S5 Data. Niger minimal neonatal dataset.**
(XLSX)

**S6 Data. Niger minimal 1–11-month-olds dataset.**
(XLSX)

**S7 Data. Tanzania minimal neonatal dataset.**
(XLSX)

**S8 Data. Tanzania minimal 1–11-month-olds dataset.**
(XLSX)

**S9 Data. Mozambique minimal neonatal dataset.**
(XLSX)

**S1 Metadata. Metadata for S1 Data-S9 Data.**
(DOCX)

**S1 Checklist. Inclusivity in global research.**
(DOCX)

## Acknowledgments

We thank Population Services International for providing the full birth history dataset that identified the deaths examined by the Cameroon VASA study, and the National Institute of Statistics of Cameroon for conducting the fieldwork to collect the Cameroon VASA data. In Nigeria, Malawi, Niger and Tanzania the national statistics office both conducted the platform household survey including a full birth history and conducted the fieldwork to collect the VASA data. We thank each of these offices, including the Nigeria National Population Commission, the National Statistics Office of Malawi, the National Statistics Institute of Niger, and the Tanzania National Bureau of Statistics and the Office of the Chief Government Statistician – Zanzibar. We also thank the National Institute of Health of Mozambique, which identified the deaths and conducted the VASA study in Mozambique under the auspices of the Countrywide Mortality Surveillance for Action (COMSA) project; and the Institute for Global Health & Development, Aga Khan University, which conducted a mortality survey with VASA as part of the Pakistan National Nutrition Survey. All seven VASA studies were conducted in collaboration with the Johns Hopkins Bloomberg School of Public Health. Alain K Koffi of Johns Hopkins assisted in the implementation of each of the African VASA studies.

## Author contributions

**Conceptualization:** Henry D Kalter, Robert E. Black.

**Formal analysis:** Henry D Kalter, Jamie Perin.

**Funding acquisition:** Robert E. Black.

**Investigation:** Zulfiqar A. Bhutta, Inuwa B. Jalingo, Paul R. Libite, Abdou Maina, Tiope Mleme, Mlemba A. Kamwe, Ivalda Macicame.

**Methodology:** Henry D Kalter, Jamie Perin.

**Project administration:** Zulfiqar A. Bhutta, Inuwa B. Jalingo, Paul R. Libite, Abdou Maina, Tiope Mleme, Mlemba A. Kamwe, Ivalda Macicame.

**Resources:** Henry D Kalter.

**Software:** Henry D Kalter, Jamie Perin.

**Supervision:** Henry D Kalter.

**Validation:** Jamie Perin.

**Visualization:** Henry D Kalter, Jamie Perin.

**Writing – original draft:** Henry D Kalter.

**Writing – review & editing:** Henry D Kalter, Jamie Perin, Zulfiqar A. Bhutta, Inuwa B. Jalingo, Paul R. Libite, Abdou Maina, Tiope Mleme, Mlemba A. Kamwe, Ivalda Macicame, Robert E. Black.

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
