## [Decision Letter · Decision Letter 0]

9 Sep 2025

PGPH-D-25-01281

Development of an effective illness severity measure and assessment of the impact of perceived illness severity on formal careseeking for fatal illnesses of neonates and infants in six sub-Saharan Africa countries and Pakistan.

Dear Dr. Kalter,

Thank you for submitting your manuscript to PLOS Global Public Health. After careful consideration, we feel that it has merit but does not fully meet PLOS Global Public Health’s publication criteria as it currently stands. Therefore, we invite you to submit a revised version of the manuscript that addresses the points raised during the review process.

The manuscript has been evaluated by one reviewer, and their comments are available below.

The reviewer has raised concerns that need attention. They request additional information including the need to clearly define the outcomes of mild, moderate, and severe illness; a clear description of when these interviews were conducted, and how the 2-sign scoring system was developed. Could you please revise the manuscript to carefully address the concerns raised?

Please note that we have only been able to secure a single reviewer to assess your manuscript. We are issuing a decision on your manuscript at this point to prevent further delays in the evaluation of your manuscript. Please be aware that the editor who handles your revised manuscript might find it necessary to invite additional reviewers to assess this work once the revised manuscript is submitted. However, we will aim to proceed on the basis of this single review if possible.

We look forward to receiving your revised manuscript.

Kind regards,

Katrien G. Janin, PhD

Staff Editor

Journal Requirements:

Please include a complete copy of PLOS’ questionnaire on inclusivity in global research in your revised manuscript. Our policy for research in this area aims to improve transparency in the reporting of research performed outside of researchers’ own country or community. The policy applies to researchers who have travelled to a different country to conduct research, research with Indigenous populations or their lands, and research on cultural artefacts. The questionnaire can also be requested at the journal’s discretion for any other submissions, even if these conditions are not met. Please find more information on the policy and a link to download a blank copy of the questionnaire here: https://journals.plos.org/globalpublichealth/s/best-practices-in-research-reporting. Please upload a completed version of your questionnaire as Supporting Information when you resubmit your manuscript.

2. Please upload all main figures as separate Figure files in .tif or .eps format only. For more information about how to convert and format your figure files please see our guidelines: LINK

Additional Editor Comments (if provided):

Reviewer #1:

Reviewers' comments:

Reviewer's Responses to Questions

**Comments to the Author**

1. Does this manuscript meet PLOS Global Public Health’s publication criteria? Is the manuscript technically sound, and do the data support the conclusions? The manuscript must describe methodologically and ethically rigorous research with conclusions that are appropriately drawn based on the data presented.

Reviewer #1: Yes

2. Has the statistical analysis been performed appropriately and rigorously?

Reviewer #1: Yes

3. Have the authors made all data underlying the findings in their manuscript fully available (please refer to the Data Availability Statement at the start of the manuscript PDF file)?

Reviewer #1: No

4. Is the manuscript presented in an intelligible fashion and written in standard English?

Reviewer #1: Yes

5. Review Comments to the Author

Reviewer #1: The authors report what appears to be a retrospective cross-sectional study to develop scoring systems to identify mild, moderate, and severe illness among young children who died using data from VASAs conducted in several sub-Saharan African and Asian countries. The strengths of this study are the multi-country nature, the extensive analyses, and the addressing of an important question (i.e., how can we best support families’ decision to seek care at an appropriate time in the course of a child’s illness).

Despite the article’s strengths, there are several weaknesses that I think should be addressed. Some of these major weaknesses include the need to clearly define the outcomes of mild, moderate, and severe illness. As written, it seems somewhat circular as caregivers seem to report these based on their understanding. Second, there needs to be a clear description of when these interviews were conducted relative to the child’s death and the events they reported about. Recall bias could potentially be huge for such details on specific days of illness if these interviews were conducted as a census planned for several years apart, which obviously wouldn’t coincide with the death of a child. Third, a better description of how the 2-sign scoring system was developed needs to be clarified. How were these features selected? Lastly, in terms of major weaknesses, as all children included here died, I’m not sure how helpful “mild and moderate” illness descriptions would be as all children actually had severe illness if they died.

I appreciate the thoughtful representation of authors from each study site. That said, it struck me as odd that in this multi-national study that included primary data collected in Tanzania did not have a Tanzanian co-author. Might the authors explain why?

More specific comments by section are below. I hope the authors find these helpful as they strengthen the reporting of their important work.

Data Availability:

-Please revise this. It does not look like “All data underlying the findings described in the paper are in the main body of the paper and in the Supporting Information file.” I defer to the journal, but I believe they are interested in having access to raw data should replication of analyses be done in the future.

Abstract:

-The authors state, “VA and VASA…ask about severe symptoms such as severe cough or inability to suckle and possibly mild or moderate symptoms…without examining progression of the child’s condition” and then state they used the very data source (i.e., VASA) for their analyses that examine “the relationship of caregivers’ perception of their child’s condition at illness onset to the child’s age and seeking formal health care”, which implies some temporality. The authors then conclude “Child mortality studies should assess severity level and caregivers’ response at various illness stages”, but how can this be a conclusion if the VASA does not assess temporality as the authors state earlier in the Abstract. Can the authors please clarify? The Introduction states that the VASA includes temporality, but the Abstract reads otherwise.

-It would be immensely helpful to describe the outcome of “mild, moderate and severe illness” better in the Abstract. This description appears to be all encompassing making it hard to understand the utility of a 2-sign method.

-I would suggest including more objective data in the Abstract to delineate the discriminatory value of these tools.

-Line 38: I would suggest avoidance of any causative language such as “this was due to” in this type of study.

Methods:

-Please include a study design.

-I appreciate the description of how the VASA was conducted differently in the different countries. That said, if some of these data were collected in a census, might the authors comment on when the interviews were conducted in relation to the death? This seems critically important as recall bias would certainly be an issue, especially when talking about nuanced temporal details about an exceptionally traumatic event.

-As above, I am not sure I entirely understand how illness severity could be classified given that the data source is one that exclusively included cases of mortality, which by default was preceded by or associated with severe illness.

-Minor comment, but lines 152 and 153 say [reference] instead of including the reference.

-Again, it’s not clear to me how the 2 signs were selected as the most optimal to identify illness severity. Please include some description of this approach.

Results:

-Please include a paragraph explaining the demographics of the included cases. I don’t see basic statistics like the number of cases, where they came from, sex, age., etc.

-What kind of care was sought as the outcome?

-I appreciate the inclusion of Tables 1 and 2 that shows site-specific categorization, but isn’t it important to also show overall performance of these approaches by pooling data across all sites?

-Line 238: Consider rewording this. As is, it seems that caregivers were less likely to seek care for more severe illness. Is that what is meant?

-Table 4 is helpful, but I suspect the authors are not powered for this type of analysis for some countries (e.g., Tanzania and Cameroon). Might the authors defend their sample size and power the this regression analysis in the Methods?

-Again, I’m not sure why the authors chose not to conduct pooled analyses across the sites.

Discussion:

-In general, this is well-written, but I think it would benefit from more of an explanation of how the authors think these scoring systems could be practically used, especially since all cases died here so they all eventually had severe illness.

6. PLOS authors have the option to publish the peer review history of their article (what does this mean?). If published, this will include your full peer review and any attached files.

**Do you want your identity to be public for this peer review?** For information about this choice, including consent withdrawal, please see our Privacy Policy.

Reviewer #1: No

---

## [Decision Letter · Decision Letter 1]

23 Mar 2026

PGPH-D-25-01281R1

Development of an effective illness severity measure and assessment of the impact of perceived illness severity on formal careseeking for fatal illnesses of neonates and infants in six sub-Saharan Africa countries and Pakistan.

Dear Dr. Kalter,

Thank you for submitting your manuscript to PLOS Global Public Health. After careful consideration, we feel that it has merit but does not fully meet PLOS Global Public Health’s publication criteria as it currently stands. Therefore, we invite you to submit a revised version of the manuscript that addresses the points raised during the review process.

We look forward to receiving your revised manuscript.

Kind regards,

Emma Clarke-Deelder

Academic Editor

Journal Requirements:

Additional Editor Comments (if provided):

Reviewers' comments:

Reviewer's Responses to Questions

**Comments to the Author**

1. If the authors have adequately addressed your comments raised in a previous round of review and you feel that this manuscript is now acceptable for publication, you may indicate that here to bypass the “Comments to the Author” section, enter your conflict of interest statement in the “Confidential to Editor” section, and submit your "Accept" recommendation.

Reviewer #1: (No Response)

2. Does this manuscript meet PLOS Global Public Health’s publication criteria? Is the manuscript technically sound, and do the data support the conclusions? The manuscript must describe methodologically and ethically rigorous research with conclusions that are appropriately drawn based on the data presented.

Reviewer #1: Yes

3. Has the statistical analysis been performed appropriately and rigorously?

Reviewer #1: Yes

4. Have the authors made all data underlying the findings in their manuscript fully available (please refer to the Data Availability Statement at the start of the manuscript PDF file)?

Reviewer #1: Yes

5. Is the manuscript presented in an intelligible fashion and written in standard English?

Reviewer #1: Yes

6. Review Comments to the Author

Reviewer #1: I appreciate the authors’ responsiveness to the peer review process. I think the manuscript reads much better at this point but I do have one concern that remains.

I’m not certain that the authors have adequately acknowledged the potential for recall bias. I had pointed out previously a concern about a need for “a clear description of when these interviews were conducted relative to the child’s death and the events they reported about.” The authors largely replied by including the years that the surveys were conducted in each country and simply added the following to the Limitations, “Being interview-based, for the most part conducted from one to five years after the death, the findings are also subject to the possible influence of faulty recall and biased reporting of events.” Surely the authors agree that an interview conducted as much as five years after a death may have some reliability concerns when asking caregivers about the “activity level and feeding behavior” “at the onset of the fatal illness”. I have two suggestions to be more explicit about this. Could the authors include some data on the timing of the interview in relation to the time of the death? Might they acknowledge this timeframe between death and the interview a bit more explicitly in the Methods?

7. PLOS authors have the option to publish the peer review history of their article (what does this mean?). If published, this will include your full peer review and any attached files.

**Do you want your identity to be public for this peer review?** For information about this choice, including consent withdrawal, please see our Privacy Policy.

Reviewer #1: No

 Figure Resubmissions:

---

## [Editor Report · Decision Letter 2]

27 Apr 2026

Development of an effective illness severity measure and assessment of the impact of perceived illness severity on formal careseeking for fatal illnesses of neonates and infants in six sub-Saharan Africa countries and Pakistan.

PGPH-D-25-01281R2

Dear Dr. Kalter,

We are pleased to inform you that your manuscript 'Development of an effective illness severity measure and assessment of the impact of perceived illness severity on formal careseeking for fatal illnesses of neonates and infants in six sub-Saharan Africa countries and Pakistan.' has been provisionally accepted for publication in PLOS Global Public Health.

Best regards,

Emma Clarke-Deelder

Academic Editor